

# A chemical transport model study of plume rise and particle size distribution for the Athabasca oil sands

Ayodeji Akingunola[1], Paul. A. Makar[1], Junhua Zhang[1], Andrea Darlington[2], Shao-Meng Li[2], Mark Gordon[3], Michael D. Moran[1], Qiong Zheng[1]

[1]Modelling and Integration Section, Air Quality Research Division, Environment and Climate Change Canada
[2] Processes Research Section, Air Quality Research Division, Environment and Climate Change Canada
[3]Centre for Research In Earth And Space Engineering, York University, Toronto Canada

*Correspondence to*: Ayodeji Akingunola (Ayodeji.akingunola@canada.ca),  Paul Makar (paul.makar@canada.ca)

**Abstract.**

We evaluate four high-resolution model simulations of pollutant emissions, chemical transformation and downwind transport for the Athabasca oil sands using the Global Environmental Multiscale – Modelling Air-quality and Chemistry (GEM-MACH) model using surface monitoring network and aircraft observations of multiple pollutants, for simulations spanning a time period corresponding to an aircraft measurement campaign in the region in summer 2013. We have focussed here on the impact of different representations of the model's aerosol size distribution and plume-rise parameterization on model results. The use of a more finely resolved representation of the aerosol size distribution was found to have a significant impact on model performance, reducing the magnitude of the original surface $PM_{2.5}$ negative biases by 32%.

We compared model predictions of $SO_2$, $NO_2$, and speciated particulate matter concentrations from simulations employing the commonly-used Briggs (1984) plume-rise algorithms to redistribute emissions from large stacks with stack plume observations. As in our companion paper (Gordon *et al.*, 2018), we found these algorithms resulted in under-predictions of plume rise, with 39 to 60% of predicted plume heights falling below half of the observed plume heights. However, we found here that a layered buoyancy approach for stable to neutral atmospheres, coupled with the assumption of free rise in convectively unstable atmospheres, resulted in much better model performance, both for atmospheric constituent concentrations and the predicted height of the plumes. Persistent issues with over-fumigation of plumes in the model were linked to positive biases in the predicted temperatures between the surface and 1km elevation. These in turn may lead to overestimates of near-surface diffusivity, resulting in excessive fumigation.



# 1 Introduction

Forecast ensembles of regional air-quality models tend to have relatively poor performance in their predictions of sulphur dioxide ($SO_2$), with normalized mean biases in the range +/-40%, Pearson's correlation coefficients (R) of less than 0.21, and normalized mean errors of more than 75% (Makar *et al.*, 2015b). These scores may be contrasted with those for atmospheric ozone ($O_3$) of +/- 13% for normalized mean bias, R more than 0.6, and normalized mean errors less than 37%. $SO_2$ is a primary emitted pollutant (it is not created by chemistry), with the majority of anthropogenic $SO_2$ emissions coming from large smokestacks (Zhang *et al.*, 2018). In North America, such "major point sources" are often outfitted with Continuous Emissions Monitoring System (CEMS) instrumentation, which provides accurate hourly estimates of the emitted mass of $SO_2$, as well as estimates of parameters that govern the buoyancy- or momentum-driven rise of the resulting plumes, such as the temperature of the emissions, and their volume flow rate (volume emitted / unit time). Anthropogenic $SO_2$ emissions are the main source of most atmospheric sulphur deposition (reacting in the gas-phase with the OH radical to create sulphuric acid, and in cloud water and rain via aqueous chemistry to create bisulphate and sulphate ions). The poor performance of $SO_2$ predictions in air-quality models is therefore a matter of concern, and drives the need to better understand its causes. Some of the potential reasons for this poor performance include (in-)accuracy of the; (i) emissions information (less likely in cases where CEMS data are available), (ii) plume rise parameterization algorithms (which describe the vertical redistribution of the emitted mass according to the stack parameters and meteorological conditions: e.g., Briggs (1984), (iii) forecast meteorological variables used in calculating plume rise, and (iv) $SO_2$ loss processes, such as oxidation (as noted above) and the deposition algorithms and meteorological inputs used for calculating the $SO_2$ deposition rate. Furthermore, a combination of these factors may drive the relative difference in model performance between $SO_2$ and $O_3$; we note, for example, that tropospheric $O_3$ is a secondary pollutant (driven by chemical formation and loss rather than direct emissions of ozone), and hence will be more spatially homogeneous than $SO_2$, with the implication that forecast accuracy for very local conditions will play more of a role in setting the ambient concentrations of $SO_2$ than $O_3$.

The prevalence of CEMS for $SO_2$ observations in both Canada and the U.S. (https://www.epa.gov/emc/emc-continuous-emission-monitoring-systems) implies that the CEMS-derived emissions inputs available for model simulations will be well characterized. However, reporting requirements vary between the countries. In Canada, emitting facilities are required to report estimates of their total annual emissions, as well as typical stack parameters, to the federal National Pollutant Release Inventory (NPRI, 2018), although individual Canadian provinces may require more detailed reporting. In the U.S., CEMS $SO_2$ data are reported at the national level to the U.S. EPA (EPA, 2018(a,b)). In both countries, estimates of the typical stack volume flow rate (and/or the stack exit flow velocity) and effluent stack exit temperatures are reported and used for modelling, instead of hourly estimates recorded by CEMS. In the Canadian province of Alberta, regulatory reporting requirements include CEMS hourly observations of $SO_2$ and $NO_2$ emissions from selected large stacks, as well as hourly information on the stack effluent temperature and volume flow rate.



In our companion paper (Gordon *et al.*, 2018) we note that past and current regional air-quality transport models (Im *et al.*, 2015; Byun and Ching, 1999; Holmes *et al.,* 2006; Emery *et al.*, 2010) and emissions processing models (CMAS, 2017; Bieser *et al.*, 2011) describe the buoyancy- and/or momentum-driven vertical redistribution of emitted mass from stacks using variations on the work of Briggs (1969, 1975, 1984). In the latter work, observations of plume rise,  stack parameter information, and meteorological conditions were used to generate parameterizations, linking these data to the height gained by the centerline of atmospheric plumes (the plume height), as well as the vertical extent of the bulk of the emitted mass about that centerline.  However, subsequent early evaluations of the accuracy of these parameterizations (cf. VDI, 1985) have had mixed results, including parameterization estimates averaging 50% higher than observations (Giebel, 1979); within 12 and 50% of observations (Ritmann, 1982); 30% higher than observations (England *et al.,* 1976); 50% higher than observations (Hamilton, 1967).  Recent studies using Reynolds averaged Navier-Stokes and large eddy simulation (RAND-LES) modelling have shown that the integral model of Briggs overestimates the plume rise and its overestimation error increases as the role of atmospheric turbulence increases (Ashrafi *et al.,* 2017), and underestimates of plume rise, inferred from excessively high predicted surface concentrations (Webster and Thompson, 2002).  Our companion paper, making use of different sources of meteorological observations, CEMS data, and aircraft observations of $SO_2$ plumes from multiple sources over a 29-day period, found that the Briggs (1984) plume rise parameterization tended to underpredict plume heights in the vicinity of the multiple large $SO_2$ emissions sources in the Canadian Athabasca oil sands, with 34 to 52% of the parameterized heights falling below half of the observed height, compared to 0 to 11% of predicted plume heights being above twice the observed height.

These underpredictions of plume rise are a potential source of concern, given that they imply that the underlying algorithms will bias $SO_2$ towards lower elevations.  This will lead to more local rather than long-range sulphur deposition.  Sulpher deposition is the focus of other work examining acidifying deposition associated with emissions sources in Alberta (Makar *et al.,* 2018).

The work reported here has four main foci, driven by the need to evaluate and if possible improve the performance of both the algorithms governing plume rise and our air-quality model (Global Environmental Multiscale – Modelling Air-quality and Chemistry; GEM-MACH) which employs those algorithms.  The main objectives of this study include: (1) an evaluation of the impacts of the plume rise algorithms on model performance, with the introduction of a new approach to calculate plume rise being compared to the standard Briggs (1984) approach; (2) estimation of the impact of hourly major point stack information on model results; and (3) an overall evaluation of the model performance using different configurations for the representation of plume rise and particle size distributions.



## 2 Model Description

### 2.1 Model Overview

Global Environmental Multiscale – Modelling Air-quality and Chemistry (GEM-MACH) is Environment and Climate Change Canada's comprehensive online air quality and chemical transport modelling system, currently in its second major

revision. The model consists of an atmospheric chemistry module (Moran *et al.,* 2010), tightly coupled with the dynamical core and residing within the physics module of the Global Environmental Multiscale (GEM) weather forecast model (Cote *et al.,* 1998).  Emissions for the model are provided using an emissions processing system based on Sparse Matrix Operator Kernel Emissions (SMOKE, Coats, 1996, https://www.cmascenter.org/smoke). GEM-MACHv2 is a multiscale model, designed and exercised in a wide range of scales, from global chemical transport modelling, to regional air quality modelling

with direct and indirect feedbacks between chemistry and meteorology (Makar *et al.,* 2015a,b), and urban scale air quality modelling (Stroud, 2016).  The chemical processes represented in the model regional air-quality prediction system includes as its main components the ADOM-II mechanism gas-phase chemistry mechanism with 42-species (Lurmann *et al.,* 1986; Stroud, 2008); the Canadian Aerosol Module (Gong et al., 2003a,b), - a size-resolved, sectional approach with either 2- or 12-size bins, multi-component, aerosol microphysics module , including process representation for particle nucleation,

condensation, and coagulation;as-phase deposition based on the work of Jarvis (1976), Wesely *et al*. (1989), and  Zhang *et al*. (2002, 2003); and particle deposition (Zhang *et al*., 2001).  Additional aerosol processes include cloud scavenging, and in-cloud aqueous phase chemistry (Gong *et al.,* 2006), as well as equilibrium inorganic gas-aerosol partitioning (HETV scheme; Makar *et al*., 2003).  Eight aerosol species are included in GEM-MACH:  particle sulfate, nitrate, ammonium, primary organic carbon, secondary organic carbon, elemental (aka "black") carbon, sea-salt and crustal material. The model

also features experimental options for feedback between weather and air-quality in 12-bin mode (Makar *et al.*,2015a,b). More detailed descriptions of GEM-MACH may be found in Makar *et al.,* 2015(a,b) and Im *et al*., 2015(a,b).  We discuss elsewhere in this special issue the use of GEM-MACH for acid deposition estimates (Makar *et al*., 2018), bi-directional fluxes of ammonia to the boreal forest (Whaley *et al.,* 2018), the impact of updated emissions of volatile organic compounds and organic particulate matter (Zhang *et al.,* 2018) on model performance for these species (Stroud *et al*., 2018).

### 2.2 Model Setup and Configurations

A 2-bin simulation of GEM-MACH running in a nested configuration from a North American 10km resolution forecast to a 2.5km Alberta/Saskatchewan domain has been in continuous experimental forecast mode since October 2012, and this configuration is also used for operational forecasts by Environment and Climate Change Canada.  While the 2-bin simulation

reduces computational processing time by 25% in the current version of GEM-MACH, we investigate here the effect of this configuration on model accuracy relative to observations, employing the GEM-MACHv2 model in the Oil Sands 2.5-km nested system using the more detailed 12-bin aerosol size distribution configuration.  We have carried out a set of retrospective simulations targeting the JOSM (the Governments of Canada and Alberta Joint Oil Sands Monitoring program)



summer 2013 intensive campaign period (JOSM, 2011). The outer 10-km horizontal resolution domain, which covers most of continental Canada and United States, was configured with 82 vertical levels and a 5-min physics/15-min chemistry time step, with the chemical boundary and initial conditions provided by MOZART-4 climatology (Emmons *et al.,* 2010), and meteorological boundary/initial conditions provided by the GEM's Regional Deterministic Prediction System (RDPS, Caron

*et al.,* 2014). The RDPS itself was driven by data-assimilated meteorological analyses. The RDPS was also used to drive a 2.5-km horizontal resolution regional weather-only simulation, using a modified GEM High Resolution Deterministic Prediction System configuration (HRDPS, Charron *et al.,* 2012). Both the modified HRDPS and the 10-km resolution GEM-MACH produced 36-hour simulations, the last 24 hours of which were used to provide the respective meteorological and chemical boundary conditions for a 24-hour GEM-MACH 2.5-km resolution simulation (which was configured with 64

vertical levels, and 1-min physics/2 min-minute chemistry model time steps). The use of the HRDPS in this fashion allowed each GEM-MACH 2.5km simulation to commence from a "spun-up" state for its cloud variables. For the chemical species, the last hour of each 24-hour simulation was used to provide initial conditions for the subsequent GEM-MACH simulation. This provided continuity of the chemical fields across subsequent 24-hour simulations. The GEM-MACH 10-km simulation and the HRDPS 2.5-km simulations, updated every 24 hours, provided ongoing boundary conditions and hence continuity

with the meteorological analysis, thus preventing the high resolution meteorology from drifting chaotically from the analyses. The GEM-MACH 10km, HRDPS, and GEM-MACH 2.5km domains are shown in Figure 1. The retrospective simulations were carried out for the period August 1$^{st}$, 2013 to September 10$^{th}$, 2013, with the first 7 days results discarded as model spin-up time.





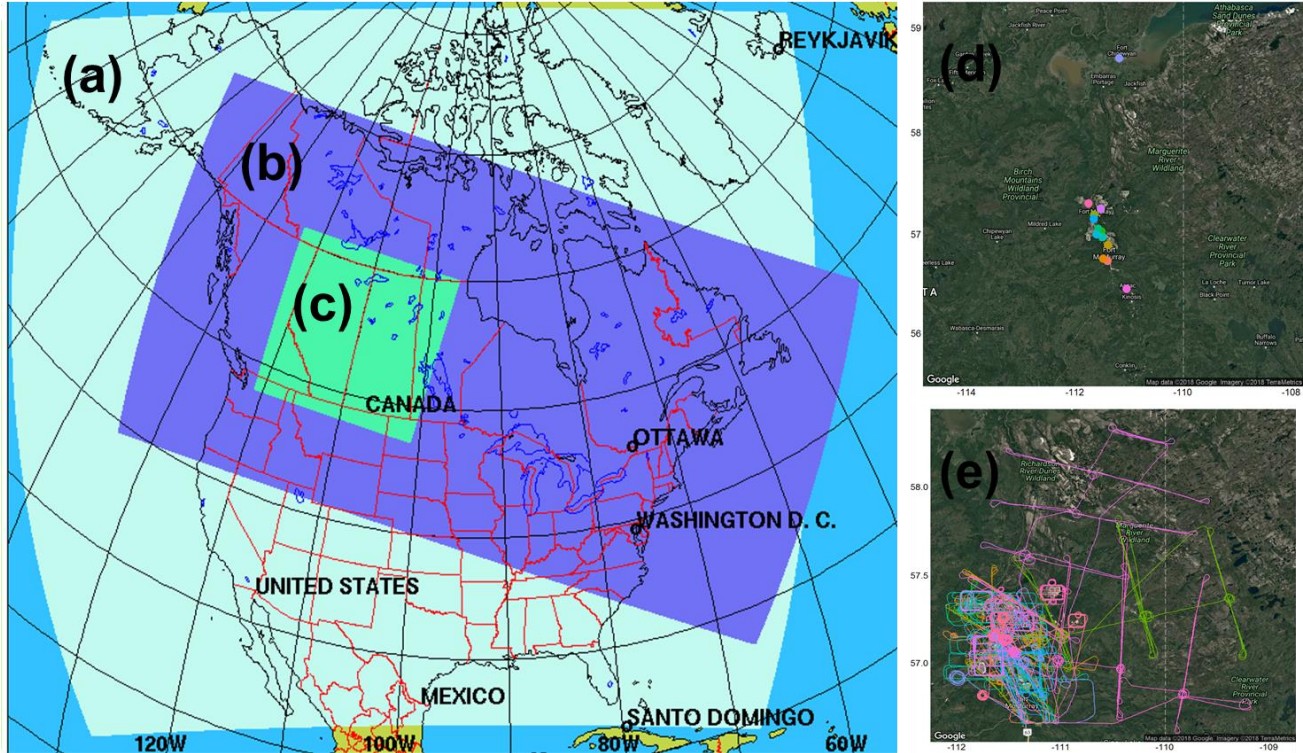

Figure 1: Schematic diagram showing the model simulation domains in the nested 2.5km resolution setup. (a) Light blue outermost domain: GEM-MACH 10km resolution North American forecast. (b) Dark blue domain: HRDPS 2.5km weather forecast. (c) Green innermost domain: GEM-MACH 2.5km forecast. (d) GoogleEarth-referenced image showing the locations of the surface observations used in the study are shown in colored dots. (e) GoogleEarth-referenced image showing all 22 flight paths covered during the JOSM 2013 flight campaign.

The emissions used in our simulations were processed from inventory data from different sources, including the Canadian National Pollutant Release Inventory (NPRI) and Air Pollutant Emissions Inventory (APEI) data for 2013, within-facility specific 2010 data from the Cumulative Environmental Management Association (CEMA), and hourly Continuous Emissions Monitoring observations for hourly major point emissions of $SO_2$ and $NO_2$ for the province of Alberta (Alberta Environment and Parks). The latter sources account for 77% and 43% of total $SO_2$ and $NO_x$ emissions, respectively, from all NPRI point sources in Alberta, and 99% and 39% respectively for sources of these compounds solely within the Athabasca oil sands area (Zhang *et al*., 2018). The same set of emissions was used for all the simulation scenarios carried out for this study. The emissions set is discussed in detail in Zhang *et al* (2016, 2018). In the emissions processing, aerosols were chemically speciated for the 12-bin size distribution; the resulting emissions files were summed to the 2-bin distribution for the 2-bin simulations discussed below.

For the purpose of this study we have carried out 4 sets of model simulations, in order to evaluate the impact of (operational) 2-bin versus 12-bin aerosol size distribution, and of different algorithms for plume rise, on model performance.



### 2.2.1 2-bin versus 12-bin model scenarios

Gong *et al. (*2003) showed that a 12-bin sectional model is sufficient to accurately predict both aerosol number concentration and mass size distributions for most prevalent atmospheric conditions. However, because of the high computational cost and the requirement for a fast turn-around demanded in operational systems, the operational forecast configuration of GEM-MACH employs a 2-bin aerosol size distribution, with sub-binning used for those aerosol microphysical processes requiring more detailed aerosol sizing, such as nucleation (Moran *et al.*, 2010). The 12-bin configuration has been used for research purposes such as investigating aerosol–weather feedbacks (Makar *et al.*, 2015a,b). Here, both aerosol size distributions were used for 10-km and 2.5-km resolution nested simulations. The first two of our simulations are thus referred to as "2-bin" and "12-bin", and both make use of the original plume rise algorithms, employed in GEM-MACH (and described below). These simulations were compared to determine the relative impact of the more detailed size distribution on model performance relative to observations.

### 2.2.2 Plume rise algorithms: two alternative approaches

As noted earlier, the set of empirical formulations and algorithms developed by Briggs (1984) for evaluating the plume rise height of major point source emissions has been the basis of plume rise calculations in several chemistry transport models such as GEM-MACH (Moran *et al.*, 2010) and CMAQ/CMAx (Byun and Schere, 2006), as well as in regulatory air dispersion models such as AEROPOL (Kaasik and Kimmel, 2003) and CALPUFF (Levy *et al.*, 2002). However, the details of how Brigg's algorithms were implemented may vary – we therefore provide the details of the GEM-MACH implementation, below. We follow with a revised plume-rise calculation procedure which in our subsequent evaluation is demonstrated to provide a more accurate estimation of final plume height. Our "12-bin" simulation noted above makes use of the original algorithm, while we refer to the revised algorithm as "Plume Rise" in our subsequent discussion.

The original implementation of the plume rise algorithm in GEM-MACH is based on the set of equations in Briggs (1984) which calculate the plume rise height above the top of the emitting stack, $\Delta h$, based on the atmospheric turbulence characteristics at the stack location. The formulae rely on a local estimation of the state of the atmosphere in the vertical at that location; the atmospheric stability, temperature gradients, and resulting formulae for plume height are predicated on the assumption that these stack-height conditions will continue throughout the atmospheric column until the maximum plume height is reached. However, in cases of more complex atmospheric conditions, where these conditions change significantly with height, the formulae may become inaccurate.

The equations depend on atmospheric stability parameters calculated in the meteorological module of the air-quality model, and include the boundary layer height ($H$), the Monin-Obhukov length ($L$), the surface wind friction velocity ($u_*$), the atmospheric temperature ($T_a$) and its gradient ($dT_a/dz$), and the wind speed (U) at the stack height. An important parameter in





the plume rise formulations is the emitted plume's initial buoyancy flux ($F_b$), which is dependent on the stack height ($h_s$), the stack exit temperature ($T_s$), and the stack's exit volume flow rate ($V$), and is given by;

$$F_b = \frac{g}{\pi} V \frac{(T_s - T_a)}{T_s}$$  (1)

Where g is the acceleration due to gravity. The emitted plume is buoyant and rises if $T_s > T_a$; $F_b$ is set to zero if $T_s < T_a$. If the stack height is within the predicted boundary layer depth ($h_s < h$), the plume rise is calculated based on the stability regimes

at the stack height model level by the following equations:

For unstable conditions ($-0.25\, h_s < L < 0$),

$$\Delta h = \min\left[3\left(\frac{F_b}{U}\right)^{\frac{3}{5}} H_*^{-\frac{2}{5}}, \quad 30\left(\frac{F_b}{U}\right)^{\frac{3}{5}}\right].$$  (2)

For stable conditions ($0 < L < 2\, h_s$)

$$\Delta h = 2.6\left(\frac{F_b}{Us}\right)^{\frac{1}{3}}.$$  (3)

And for neutral conditions ($L > 2\, h_s$ and $L < -0.25\, h_s$),

$$\Delta h = \min\left[39\frac{F_b^{3/5}}{U}, \quad 1.2\left(\frac{F_b}{u_*^2 U}\right)^{3/5}\left(h_s + 1.3\frac{F_b}{u_* U}\right)^{2/5}\right].$$  (4)

Where $H_* = -2.5 u_*^3 / L$ is the convective scale velocity, and $s$ is the stability parameter approximated by:

$$s = \frac{g}{T_a}\left(\frac{dT_a}{dz} + \frac{g}{c_p}\right).$$  (5)

Where $dT_a/dz$ is the vertical temperature gradient between the atmospheric temperature at the top of the stack and the temperature at the top of the model layer. We note here that some air-quality model implementations make use of one or the other formula of equations (2) and (4), as opposed to the minimum chosen here. In our companion paper (Gordon *et al.*, 2018) we show that these differences have little impact on the calculated plume height.

The model also incorporates the potential for the buoyant plume to penetrate the top of the boundary layer (Hanna and Paine,

1988), which is accounted for by calculating the penetration parameter $P$ and using it to further adjust the plume rise $\Delta h$ calculated through the above formulae as:

$$P = \begin{cases} 1, & \frac{H - h_s}{\Delta h} \le 0.5 \\ 1.5 - \frac{H - h_s}{\Delta h}, & 0.5 < H \le 1.5 \\ 0, & \frac{H - h_s}{\Delta h} \ge 1.5 \end{cases}$$  (6)

Where H is the height of the boundary layer. The plume rise calculated earlier is then reset via;

$$\Delta h = min[(0.62 + 0.38P)(H - h_s), \Delta h]$$  (7)





Once the final value of the plume rise $\Delta h$ is calculated, the vertical spread of the plume and the emitted mass is then evaluated by using a common method from Briggs (1975) to specify the height of the top and bottom of the plume as;

$$h_{top} = h_s + 1.5\Delta h$$
$$h_{bottom} = h_s - 0.5\Delta h \qquad (8)$$

In GEM-MACH, the plume top is further limited to the height of the boundary layer ($H$), if the penetration $P > 0$. During

unstable conditions, the plume bottom is set to zero (the surface); that is, the plume is assumed to mix uniformly between the top of the atmosphere and the surface. We also note that the mass emitted into the plume is assumed to mix uniformly between $h_{top}$ and $h_{bottom}$; this is in contrast to the approach of Turner (1991), wherein a "top-hat" distribution centered on the value of hs was assumed, or the Gaussian distribution based on unpublished observations described in Byun and Ching (1999).

As described above, the original plume rise algorithm implemented in GEM-MACH does not account for potential changes in plume rise associated with the vertical variation in the atmospheric temperature and stability, which could be important for plume buoyancy especially during unstable conditions where the boundary layer depth could be much higher than the stack height. Similarly, changes in stability with height will affect plume rise. As reported in Gordon *et al.* (2018), when

meteorological *observations* collected at oil sands sites are used to drive equations (1) through (5), the estimated plume heights were often underestimated, with between 37 to 52 percent of calculated values being less than ½ the observed height.

However, other approaches, which take into account the variation in height associated with atmospheric conditions in the

vertical profile above the emitting stack, are available. Briggs proposed equations which would make use of changes in stability between layers and calculate the residual buoyancy flux between layers in the atmosphere – these are particularly amenable to the layered structure of atmospheric models (Briggs, 1985, equations 8.84 and 8.85). This new algorithm is similar to other layer-by-layer approaches available in CMAQ (Byun and Ching, 1999), based on the hesitant-plume algorithm described in Turner (1991) and in dispersion modelling work by Erbrink (1994). In the new algorithm (hereafter

referred to as the revised Briggs Plume rise or simply "plume rise") we utilized the model's calculated vertical profile of atmospheric temperature and wind speed to estimate the plume height as the height at which the emitted plume buoyancy flux dissipates totally. The initial plume buoyancy flux ($F_b$) at the top of the stack is calculated using equation (1) above, by using linear interpolation to evaluate the air temperature ($T_a$) and wind speed ($U$) at the stack height from the model's vertical profile. Under (locally) neutral and stable conditions, the buoyant plume is assumed to rise freely, and the residual

buoyancy flux ($F_r$), remaining after it as it crosses the next atmospheric layer is given by:





$$F_{j+1} = \begin{cases} F_j - 0.015 s_j F_{j-1}^{\frac{1}{3}}(z_{j+1}^{\frac{8}{3}} - z_j^{\frac{8}{3}}), & vertical\ plumes \\ F_j - 0.053 s_j U_j(z_{j+1}^{3} - z_j^{3}), & bent\ plumes \end{cases} \qquad (9)$$

Here, $s_j$ is the local stability parameter for a given layer, calculated using (5) and layer-specific temperature values, and $z_j$ is the plume rise height when the plume reaches the bottom of the model's j'th layer. Briggs (1984) recommended the use of *both* formulae of (9), with the formula with greatest decrease in flux being used as the final value. Briggs also noted that the transition to bent plumes happens at a relatively low height above the stack, implying that that the residual buoyancy

between layers is lost faster under windy conditions. At the stack height, $F_{j=0} = F_b$, and $= h_s$. When the residual buoyancy flux becomes negative in (9), indicating that the plume height has been surpassed, the calculation is repeated to find the value of z for which F=0; the sum of this and the layer thicknesses transitioned to this height becomes the predicted plume rise. In our companion paper (Gordon *et al.*, 2018), this approach was found to provide similar results to the original Briggs' algorithms when driven by observations. Our work here indicates that this algorithm has the potential to provide a more

accurate estimate of plume rise, subject to caveats described below.

We note that the numerical coefficients in (9); 0.015 and 0.053, stem from two parameters; the entrainment constant for vertical rise conditions (α, the entrainment coefficient for vertical plumes, nominally set to 0.08 by Briggs based on observations published in 1975 - the parameter in the first equation of (9) is a non-linear function of this α term; and β', the entrainment coefficient internal radius for bent-over plumes, set to Briggs as 0.4, though ranges from 0.45 to 0.52 were

quoted elsewhere in Briggs, 1984). The choice of these parameters are based on data which are now over 40 years old, and may present an opportunity for future improvement of this revised plume rise approach.

The above formula (9) was recommended by Briggs for conditions which are stable to neutral at the stack height. We have defined stability in this case by comparing the dry adiabatic lapse rate to the local temperature lapse rate predicted by the model at the stack height and above. Briggs (1984) provided no equivalent formula for unstable conditions at the stack

height, followed by stable profiles at higher elevations. The approach taken here has been to assume under convectively unstable conditions, the plume rises without loss of energy (that is, an assumption of zero entrainment) until the predicted temperature profile once again falls below the dry adiabatic lapse rate. Our first order approximation is thus to assume that under unstable conditions, there is minimal mixing entrainment of the rising plume with the surrounding atmosphere. This approach differs from that of Turner (1991), and the layered approach described in Byun and Ching (1999) where the

residual buoyancy flux between layers is determined using different formulae based on the model-determined local atmospheric stability.

As in the original algorithm, the plume top and plume bottom are evaluated using equation (8) after the final plume rise has been evaluated. We do not apply the penetration equations (6 and 7) since these corrections should be unnecessary in an approach making use of local changes in residual buoyancy. In our companion paper, this algorithm is referred to as the

"layered approach".



### 2.2.3 Hourly Emission Stack Temperature and Volume Flow Rate

We turn next to the available emissions data for driving the plume rise algorithms. Under Canadian federal reporting requirements to the National Pollutant Release Inventory (NPRI), annual total emissions of $SO_2$ and $NO_x$ from facilities are reported, along with a single set of stack parameters (stack height, stack diameters, average exit temperature, and average exit velocity) to represent emissions throughout the year. In addition, hourly Continuous Emissions Monitoring data from large stacks are reported to the government of Alberta. These data include the hourly mass of emissions of $SO_2$ and $NO_2$, as well as hourly estimates of the time-varying stack parameters (volume flow rates and temperatures). Our first two simulations use the "standard" annual NPRI reported stack parameters and the original plume rise algorithm for the 2-bin and 12-bin aerosol size distributions, while our second two simulations use the modified plume rise algorithm, first with the NPRI stack parameters, and second with CEMS derived hourly stack parameters. The four scenarios examined are thus:

(1) A "2-bin" simulation: NPRI stack parameters, 2-bin aerosol size distribution, and the original plume rise

(2) A "12-bin" simulation: As in (1), but employing the 12-bin aerosol size distribution. Differences between (1) and (2) thus show the impact of the aerosol size distribution on performance.

(3) A "Plume rise" simulation: employing the layered plume rise algorithm, with emissions as in (2)    Differences between (2) and (3) thus show the impact of the revised plume rise algorithm alone.

(4) An "Hourly" simulation: employing the layered plume rise algorithm, with volume flow rates and temperatures taken from the hourly CEMS data.    Differences between (3) and (4) thus show the impact of the initial buoyancy flux on the resulting plume rise, using the revised algorithm.

All of these simulations make use of the CEMS-derived mass of emitted $SO_2$ and $NO_x$.

### 3 Observations

The comparative statistics presented through this study were computed using the 'modstat' function from the openair R package (Carslaw and Ropkins, 2012), for complete pairs of valid model and observation data. Both surface monitoring network and aircraft observations have been used for model evaluation.

### 3.1 WBEA Surface Monitoring Networks

For the purpose of model evaluation, we have used hourly measurements of surface concentrations of $PM_{2.5}$, $SO_2$, $NO_2$, and $O_3$ from a network of 10 air quality monitoring stations in the province of Alberta managed by the Wood Buffalo Environmental Association (WBEA) (see Figure 1(d)). The observation data have been filtered to remove extreme single-hour measurements that are greater than 150ppbv for $SO_2$, $NO_2$, and $O_3$, and 150 µg m$^{-3}$ for $PM_{2.5}$. Observations from August 10$^{th}$, 2013 to September 10$^{th}$, 2013 were selected for comparison to the model results, to align to with the period covered by the JOSM 2013 intensive aircraft measurement campaign.



### 3.2 JOSM Summer 2013 Intensive Campaign

From August 10[th] to September 10[th], 2013, the National Research Council of Canada Convair aircraft was used as a mobile measurement platform to sample atmospheric constituents in the region of the Athabasca oil sands, with twenty-two flights taking place during the given time period (Figure 1e). These flights included flight paths designed for emission estimation,

for the study of downwind transport and chemical transformation, and for satellite validation. Emission estimation flights took place around individual facilities at multiple altitudes, with the concentration and meteorological information gathered subsequently used to estimate fluxes entering and leaving the facility, and hence estimate emissions directly from aircraft observations (Gordon *et al.*, 2015; Li *et al.*, 2017)  Transformation flights were designed to follow plumes downwind, with observations taken in cross-sections at set distances downwind perpendicular to the plume direction, in order to study

chemical transformations between point of emission and downwind receptors (Liggio *et  al.*, 2016). Satellite validation flights incorporated aircraft vertical spirals at satellite overpass times, in order to improve satellite data retrieval algorithms (Whaley *et al.*, 2018; Sheppard *et al.*, 2015). Here, we compare model predictions for our different simulations for $SO_2$, $NO_2$ and for PM1 sulfate, ammonium, and total organics to observations taken on-board the Convair using TS43, TS42 and Aerodyne Aerosol Mass Spectrometers (AMS) instruments, respectively. In order to allow for comparisons to the results

from GEM-MACHv2 2.5km oilsands model domain simulations, 10-second averages of the aircraft's positional data (latitude, longitude, elevation, and time) were created for all 22 flights. These data were in turn used to extract the corresponding time and spatial linearly interpolated model values at the model's chemistry time resolution of 2 minutes, for each of the instruments aboard the aircraft that were used for the model comparison.

### 4 Results and Discussion

We begin our evaluation by comparing the 2-bin and 12-bin particle size distribution simulations using identical emissions against Wood Buffalo Environmental Association's surface monitoring network $PM_{2.5}$ measurements.  The statistical comparison between these observations and all the 4 model scenarios is shown in Table 1, and the corresponding  histograms

of observations (blue), 2-bin model simulated values (red) and 12-bin model simulation values (purple) is shown in Figure 2. The statistics of Table 1 show that the 12-bin simulation provides an overall improvement over the 2-bin model results across all metrics.  For example, the magnitude of the negative bias has decreased by 34%, indicating that a sizeable fraction of particulate under-predictions in 2-bin simulations may be due to poor representation of particle microphysics through the use of the 2-bin distribution, despite sub-binning being used in some microphysics processes.  The largest improvement in

correlation coefficient and fraction of predictions within a factor of two also takes place going from the 2 to the 12 bin distribution. Figure 2 shows that the model simulations are biased high for particles less than 5 μm diameter, and biased low for the larger particle sizes.  However, the use of the 12-bin size distribution (purple histogram bars, Figure 2) improves the fit to the observations (blue histogram bars), in comparison to the 2-bin distribution results (red histogram bars).




The simulation with the largest number of highest scores (bold-face numbers in Table 1) is the "Plume rise" algorithm , which made use of the revised plume rise formulation, though the differences in performance between the "12-bin", "Plume rise" and "Hourly" simulations are relatively small. The latter small increment is expected, given that the observations are relatively close to the sources of primary particulate emissions, largely from surface sources of fugitive dust (see Zhang *et*

*al.,* 2018). However, an increment of $PM_{2.5}$ will be from secondary sources; about 99% of the anthropogenic $SO_2$ and $NH_3$ emissions, and about 40% of the $NO_x$ emissions in the Athabasca oil sands region originate in "major point source" stacks. The concentrations of these precursor species will therefore be influenced by the plume rise algorithm employed in model simulations, and hence secondary particulate species originating from these primary emissions may also be affected by plume rise. The small improvements in $PM_{2.5}$ associated with the revised plume rise algorithm may thus represent the impact

of secondary formation of particulate sulphate, ammonium and nitrate from $SO_2$, $NH_3$, and NOx, the latter having been influenced by the plume rise treatment. We examine this possibility using observations of $PM_1$ particle sulphate, and ammonium taken with an aerosol mass spectrometer (AMS) aboard the NRCan Convair aircraft.

The aircraft's AMS instrument measures speciated atmospheric particle concentrations for particles less than 1µm size, and therefore cannot be compared with the 2-bin model results (the two size bins are from 0 to 2.56 µm and 2.56 to 10.24 µm,

hence the smaller size bin will be biased high relative to the 1 µm size cut of the AMS). While the modelled $PM_1$ organic aerosols (OA) compared similarly to the AMS measurements for all the 3 model scenarios, the $PM_1$ sulphate and ammonium simulations with revised plume rise algorithm ("Plumerise" and "Hourly" simulations) produced better scores for most statistics than the "12-bin" simulation compared to the original plume rise algorithm. Particulate sulphate largely originates in atmospheric oxidation of $SO_2$ by the OH radical in these flights – relatively little sulphate is emitted directly, and aqueous

oxidation is largely absent due to the flights being cloud-free. Particle ammonium levels are closely linked to the sulfate through inorganic chemistry as well as being emitted by stacks in this region, and hence the ammonia results are consistent with the sulphate. The organic aerosols are at this distance downwind largely due to formation from area emissions sources of primary organic aerosol and of precursor volatile organic compounds to secondary organic aerosol formation, rather than large stack emissions, and hence are less affected by the plume rise treatment. A larger influence of plume rise on model

results is expected for $SO_2$ and $NO_2$, due to the large fraction of their emissions originating in the large stacks of the Athabasca oil sands facilities.



**Table 1**: Statistical comparison of GEM-MACH model simulation of surface PM$_{2.5}$ with measurements from the WBEA observations between August 10$^{th}$ and September 10$^{th}$, 2013. **Bold** face:  best score.  *Italics*:  second best score.

| Statistic | PM$_{2.5}$(µg/m$^3$) | | | |
|---|---|---|---|---|
| | **2-bin** | **12-bin** | **Plumerise** | **Hourly** |
| Number of complete data pair, **n** | 6815 | 6815 | 6815 | 6815 |
| Fraction of predictions within a factor of two, *FAC2* | 0.386 | 0.454 | **0.456** | *0.455* |
| Mean bias, ***MB*** | -2.623 | **-1.725** | -1.813 | *-1.807* |
| Mean Gross Error, ***MGE*** | 4.852 | 4.742 | **4.690** | *4.696* |
| Normalised mean bias, ***NMB*** | -0.39 | **-0.257** | -0.270 | -0.269 |
| Normalised mean gross error, ***NMGE*** | 0.722 | 0.705 | **0.698** | *0.699* |
| Root mean squared error, ***RMSE*** | 8.447 | 8.442 | **8.359** | *8.363* |
| Correlation coefficient, ***r*** | 0.122 | 0.151 | *0.154* | **0.155** |
| Coefficient of Efficiency, ***COE*** | -0.213 | -0.185 | **-0.172** | *-0.174* |
| Index of Agreement, ***IOA*** | 0.394 | 0.407 | **0.414** | *0.413* |

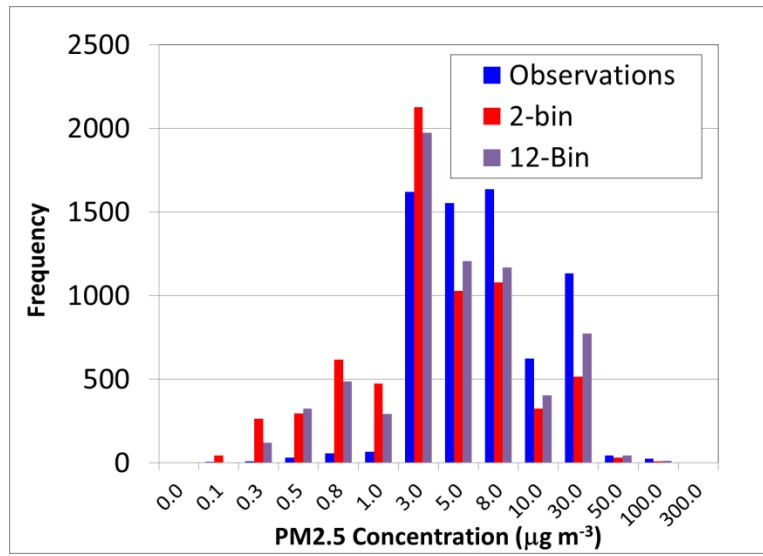

Figure 2. Histogram of surface PM2.5 using Wood Buffalo Environmental Association surface monitoring data (blue), and the 2-bin (red) and 12-bin (purple) configurations of GEM-MACH.  Both simulations make use of the original Briggs (1984) plume rise formulation.



**Table 2**: Statistical comparison of PM₁ sulphate, OA and ammonium atmospheric concentration from the aircraft AMS with the 2.5km resolution GEM-MACH simulations between August 13th and September 9th, 2013. **Bold** face: best score. *Italics*: second best score.

| Statistic | SO$_4$(µg/m$^3$) 12-bin | Plumerise | Hourly | OA(µg/m$^3$) 12-bin | Plumerise | Hourly | NH$_4$(µg/m$^3$) 12-bin | Plumerise | Hourly |
|---|---|---|---|---|---|---|---|---|---|
| n | 24523 | 24523 | 24523 | 24522 | 24522 | 24522 | 24523 | 24523 | 24523 |
| FAC2 | **0.475** | 0.467 | 0.466 | **0.138** | 0.137 | 0.137 | **0.527** | **0.527** | 0.525 |
| MB | -0.445 | -0.489 | **-0.435** | **-2.67** | -2.669 | -2.669 | **-0.014** | -0.034 | -0.023 |
| MGE | 0.964 | **0.925** | 0.959 | **2.725** | 2.727 | 2.727 | 0.272 | **0.252** | 0.261 |
| NMB | -0.397 | -0.436 | **-0.388** | -0.714 | *-0.713* | *-0.713* | **-0.051** | -0.121 | -0.081 |
| NMGE | 0.861 | **0.826** | 0.856 | **0.728** | 0.729 | 0.729 | 0.976 | **0.904** | 0.937 |
| RMSE | 4.629 | **4.592** | 4.608 | 3.773 | 3.773 | 3.773 | 1.176 | **1.124** | 1.141 |
| r | 0.148 | 0.171 | **0.175** | **0.552** | 0.548 | 0.549 | 0.149 | 0.175 | **0.178** |
| COE | 0.203 | **0.235** | 0.207 | -0.111 | -0.111 | -0.111 | 0.061 | **0.13** | 0.099 |
| IOA | 0.601 | **0.618** | 0.603 | **0.445** | 0.444 | 0.444 | 0.53 | **0.565** | 0.549 |

The performance of the three model simulations using different plume rise algorithms, for surface mixing ratios of SO$_2$ observed at WBEA stations, is shown in Figure 3. The model simulations are biased low for zero concentration levels (first bin, Figure 3(a)), are biased high from 0.0 to 0.3 ppbv, biased low from 0.3 to 1.0 ppbv (Figure 3(a)), and biased high for all concentrations above 1 ppbv (Figure 3(b,c)). These last two ranges (Figure 3(b,c)) result from surface fumigation of high

concentration plumes in the region studied. While all model simulations are biased high for these fumigating plumes, the "Plumerise" and "Hourly" simulations have a reduced bias compared to the original plume rise algorithm "12-bin". The use of the hourly stack parameters derived from Continuous Emissions Monitoring ("Hourly") has somewhat worse performance than the same plume rise algorithm driven by annual reported stack parameters ("Plumerise").

The SO$_2$ statistics of Table 3 show sometimes substantial improvements in model performance with the use of the revised

plume rise algorithms, with the mean bias being reduced by 61%, the mean gross error by 27%, the correlation coefficient increasing by 26% and the index of agreement increasing by a factor of 2.26 between the 12-bin and "Plumerise" algorithms, and the "second best" score (italics) out of the three simulations being the "Hourly" simulation employing the revised volume flow rates and stack temperatures. For NO$_2$, the "Hourly" values (employing the hourly volume flow rates and temperatures) tend to have the best scores, though the differences between "Hourly" and "Plumerise" simulations, where the

only difference in the plume treatment is in the source of data for the initial buoyancy flux, is relatively small. Both of the





primary pollutants have shown a noticeable improvement in performance with the new plume rise treatment, with the pollutant for which most emissions are from stacks ($SO_2$) having the most noticeable changes.

Ozone, in contrast, is created or destroyed through secondary chemistry over relatively longer time-spans than the transport time from the sources in this comparison (spatial scales on the order of 10's of km). Accordingly, the impact of the plume

5    rise of $NO_x$ on ozone formation is relatively minor, usually in the third decimal place (though first decimal place improvements occur for the mean bias with the use of the new plume rise algorithm).

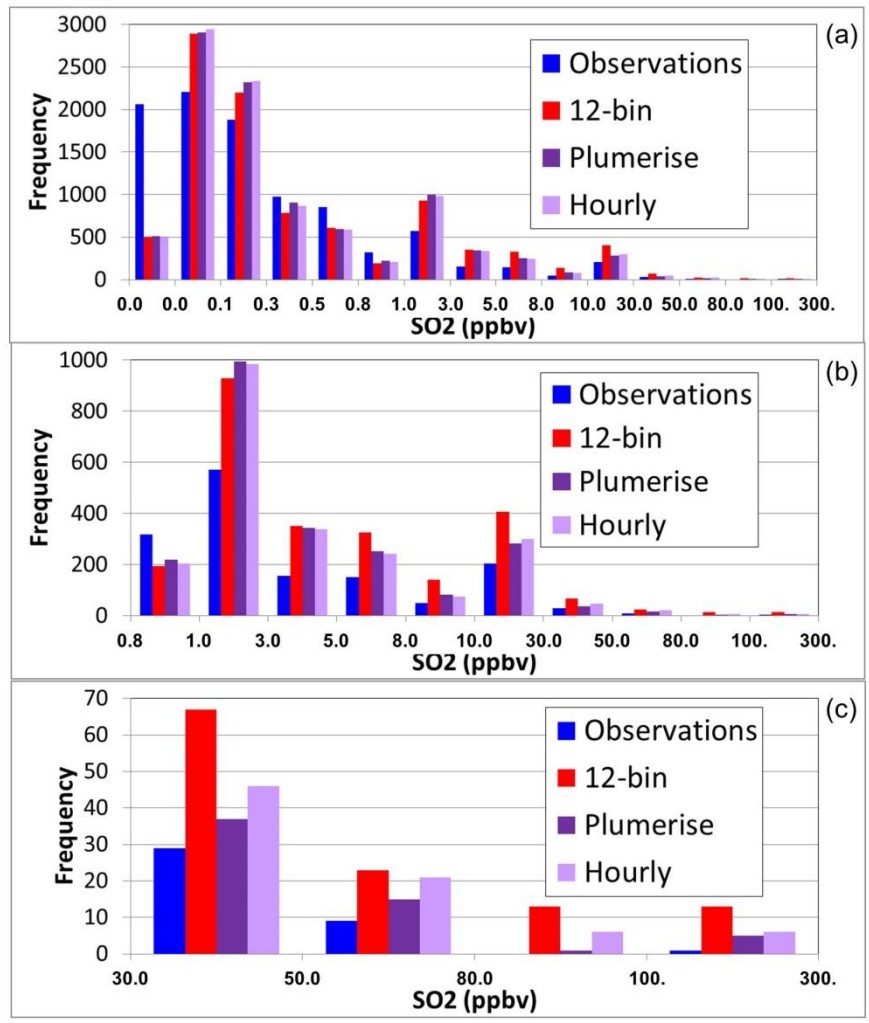

10    Figure 3: Histograms of hourly surface $SO_2$ mixing ratios, in logarithmic mixing ratio bins, observations (blue), original plume rise algorithm (red), revised plume rise algorithm (dark purple), revised plume rise algorithm driven by hourly CEMS stack data (light purple). (a) All values. (b) 0.8 to 300 ppbv. (c) 30 to 300 ppbv.



**Table 3**: Statistical comparison of $SO_2$, $NO_2$, and $O_3$ surface concentration measurements from the WBEA surface observation network with the 2.5km resolution GEM-MACH simulations between August 10[th] and September 10[th], 2013.

| Statistic | $SO_2$(ppb) | | | $NO_2$(ppb) | | | $O_3$(ppb) | | |
|---|---|---|---|---|---|---|---|---|---|
| | 12-bin | Plumerise | Hourly | 12-bin | Plumerise | Hourly | 12-bin | Plumerise | Hourly |
| **n** | 9457 | 9457 | 9457 | 6516 | 6516 | 6516 | 4384 | 4384 | 4384 |
| *FAC2* | 0.226 | **0.239** | *0.238* | 0.324 | *0.328* | **0.329** | **0.778** | *0.777* | *0.777* |
| *MB* | 1.215 | **0.476** | *0.631* | 1.006 | *0.919* | **0.910** | -0.977 | -0.907 | **-0.898** |
| *MGE* | 2.41 | **1.756** | *1.906* | 4.456 | *4.388* | **4.384** | **7.239** | 7.269 | 7.271 |
| *NMB* | 1.133 | **0.444** | 0.588 | 0.262 | *0.239* | 0.237 | *-0.051* | **-0.047** | **-0.047** |
| *NMGE* | 2.247 | **1.637** | *1.777* | 1.159 | *1.141* | **1.140** | **0.375** | 0.376 | 0.377 |
| *RMSE* | 8.99 | **6.291** | *7.018* | 7.742 | *7.656* | **7.651** | **9.690** | *9.750* | 9.771 |
| *r* | 0.179 | **0.227** | *0.218* | **0.287** | **0.287** | *0.286* | **0.617** | 0.613 | 0.612 |
| *COE* | -0.648 | **-0.201** | -0.304 | -0.264 | *-0.244* | **-0.243** | 0.220 | *0.217* | *0.217* |
| *IOA* | 0.176 | **0.399** | *0.348* | *0.368* | **0.378** | **0.378** | **0.610** | *0.609* | 0.608 |

Overall, these results suggest that (a) the revised plume rise algorithm improves the model surface performance for primary pollutants largely emitted from stack sources ($SO_2$) or for which a large proportion of the emitted mass is via stack sources ($NO_2$). Also, the impact of the hourly volume flow rates and temperatures versus typical annual values is relatively small, though it results in a degradation of performance.

Statistical comparisons of model results computed against aircraft observations for $SO_2$, $NO_2$ and $O_3$, for all the flights in the aircraft campaign are shown in the Table 4. Histograms of model performance for $SO_2$ aloft are shown in Figure 4. With the exception of more negative biases, the two sets of atmospheric $SO_2$ concentrations calculated by the new plume rise algorithm driven using annual reported stack parameters again give the best results when compared to the aircraft measurements, for all statistical measures aside from the biases (the "Plume rise" and "Hourly" simulations are biased lower than the 12-bin simulation). The variation in the statistical performance between different plume rise algorithms aloft are larger than those noted above for the surface observation comparisons, for the model scenario with "Plume rise" and "Hourly" scenarios, with the former having the best overall performance. A more substantial improvement for $NO_2$ with the revised plume rise algorithm may be seen in comparison to the surface observation evaluation, with larger decreases in the mean bias, mean gross error, root mean square error, and increases in the scores for correlation coefficient, coefficient of error and index of agreement, between the "12-bin" and "Plumerise" simulations. The results for the two simulations using the new plume rise algorithm however remain similar for $NO_2$. It should be noted as well that the model generally performs





better against the aircraft measurements than the comparisons to the surface observations across all the statistical measures for $NO_2$, reflecting the aircraft sampling a greater proportion of $NO_2$ mass originating from elevated plumes as opposed to surface sources. Similar to the surface observation comparisons, the atmospheric $O_3$ concentration calculated by the various model scenarios shows very minimal variation in the comparative statistics with the aircraft observation, with the exception

5   of a marginally better correlation coefficient ($r = 0.477$) for original plume rise scenario compared to the result ($r = 0.6947$) for the new plume rise scenarios.

Figure 4 shows the histograms comparing aircraft observations with the results of the three variations of plume rise algorithms for $SO_2$ (Figure 4(a,b)) and $NO_2$ (Figure 4(c,d)). In contrast to Figure 3, all model simulations for $SO_2$ aloft are biased *low* between mixing ratios of 0.3 and 50 ppbv, and remain biased low above 50 ppbv for the Plume rise and Hourly

10  simulations. Thus, model estimates of *surface* $SO_2$ mixing ratios (Figure 3) are biased high, while *aloft* (Figure 4(a,b)), $SO_2$ mixing are biased low. A similar, though less pronounced, pattern may be seen for $NO_2$ (Figure 4(c,d)), with model mixing ratios aloft biased low, for histogram bins between 0.1 and 10 ppbv. All versions of the model thus have a tendency to underpredict the *height* of the plumes, overestimating surface fumigation events, and underestimating occurrences when the plume remains aloft.

**Table 4**: Comparison of statistical measures of $SO_2$, $NO_2$, and $O_3$ measurements from the aircraft campaign against the 2.5km resolution GEM-MACH simulations between August 13[th] and September 10[th], 2013.

| Statistic | TS43 - SO₂(ppb) | | | TS42 - NO₂(ppb) | | | TS49 - O₃(ppb) | | |
|---|---|---|---|---|---|---|---|---|---|
| | 12-bin | Plumerise | Hourly | 12-bin | Plumerise | Hourly | 12-bin | Plumerise | Hourly |
| n | 29313 | 29313 | 29313 | 28114 | 28114 | 28114 | 29263 | 29263 | 29263 |
| FAC2 | 0.233 | **0.249** | *0.243* | *0.300* | **0.306** | **0.306** | **0.953** | *0.950* | 0.949 |
| MB | **-0.186** | -0.795 | *-0.444* | *0.097* | **-0.007** | **-0.007** | -1.729 | *-1.539* | **-1.536** |
| MGE | 4.031 | **3.438** | 3.705 | 1.482 | **1.362** | *1.366* | **8.842** | 8.915 | 8.919 |
| NMB | **-0.057** | -0.244 | *-0.136* | 0.065 | **-0.005** | **-0.005** | *-0.056* | **-0.050** | **-0.050** |
| NMGE | 1.237 | **1.055** | *1.137* | 0.988 | **0.908** | *0.911* | **0.287** | *0.290* | *0.290* |
| RMSE | 14.332 | **11.153** | *12.345* | 3.521 | **2.951** | *2.957* | **11.757** | *11.961* | 11.982 |
| r | 0.234 | **0.34** | *0.317* | 0.416 | **0.493** | *0.491* | **0.477** | 0.469 | 0.468 |
| COE | 0.142 | **0.268** | *0.211* | 0.177 | **0.244** | *0.241* | **-0.171** | *-0.180* | -0.181 |
| IOA | 0.571 | **0.634** | *0.606* | 0.589 | **0.622** | *0.621* | **0.415** | *0.410* | *0.410* |





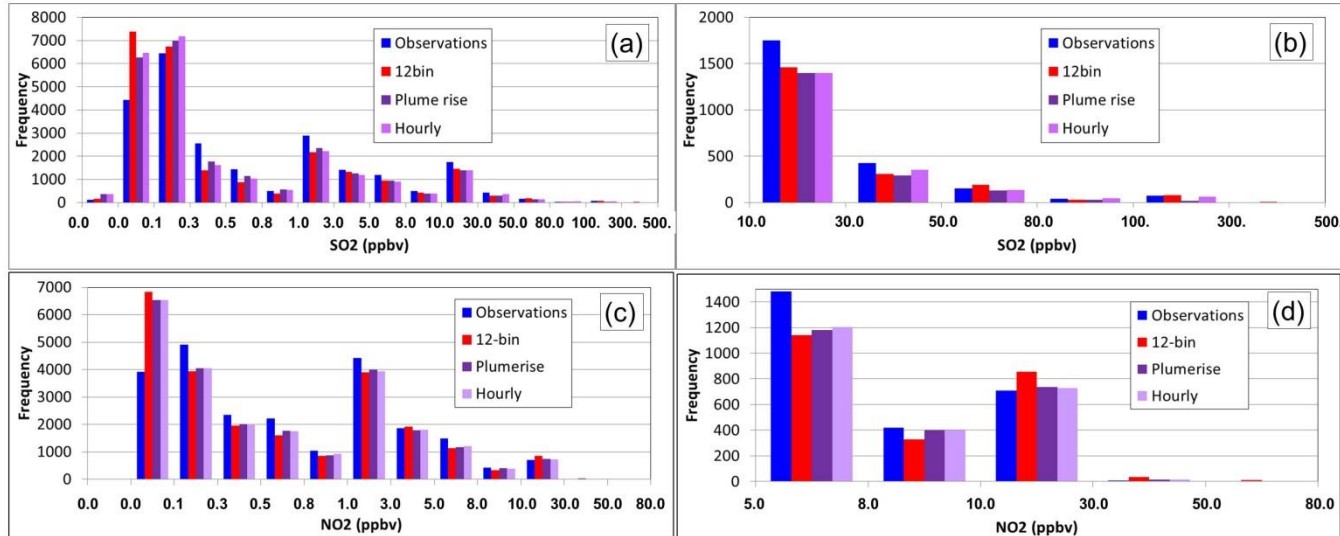

Figure 4: Histograms comparing $SO_2$ and $NO_2$ simulations mixing ratios (ppbv) with aircraft observations. (a) All $SO_2$ values. (b) Higher $SO_2$ mixing ratios. (c) All $NO_2$ values. (d) Higher $NO_2$ mixing ratios.

The results across the different simulations suggest that the overall model performance may be hampered by a tendency to place too much emitted mass close to the surface, and insufficient mass aloft. In order to determine possible causes for this behaviour, we carried out several additional analyses.

First, we examined the 12[th] flight of the observation study, which took place between 16:30 and 20:30 on Aug 24[th], as a case study to show the differences between the three simulations examining the impacts of the choice of plume rise algorithm and its input parameters. Flight 12 was an "emissions" flight, with the aircraft flying around the boundary of a single facility (Syncrude), with elevations gradually increasing in two successive sets of passes around the boundary. Data collected during flights of this nature were used to estimate emissions from the facility via calculation of the fluxes into and out of the facility from the collected data (Gordon *et al.*, 2015). During flight 12, the aircraft carried out two successive sequences circling the facility boundaries in gradual upward spirals (between 17:00 to 18:15, and 18:45 to 19:45), starting at the lowest aircraft altitude above the surface, and gradually increasing in elevation on each pass around the facility. The $SO_2$ plume was thus intersected at multiple times and multiple heights during each of these periods. Figure 5(a,b,c) depicts the model-derived $SO_2$ mixing ratio profiles at 10 second intervals interpolated from the aircraft positions as a function of time, as mixing ratio contours, for the "12-bin", "Plume rise" and "Hourly" scenarios, respectively. The aircraft locations are shown as coloured dots over-plotted on the background model mixing ratio contours. Each high mixing ratio "spike" in the panels



of Figure 5 thus represents a successive pass through the model $SO_2$ plume – the change in these plumes as a function of time may be seen by following the changes in the plume cross-sections in each panel along the x-axis timeline, from left to right. Between 17:00 and 18:15, the simulated plumes are mostly aloft. The 12-bin simulation employing the original Briggs algorithm (Figure 5(a)) begins to fumigate significantly by 17:30, with higher concentrations reaching the surface, while for the Plume rise simulation (Figure 5(b)) the plume both reaches higher elevations, and experiences significantly less fumigation. The Hourly simulation (Figure 5(c)) is intermediate between the other two simulations. In the second period (18:45 to 19:45), the fumigation behaviour becomes more pronounced for all three simulations, and once again is strongest for the 12-bin simulation (Figure 5(a), weakest for the Plume rise simulation (Figure 5(b)), and intermediate for the Hourly simulation (Figure 5(c)).





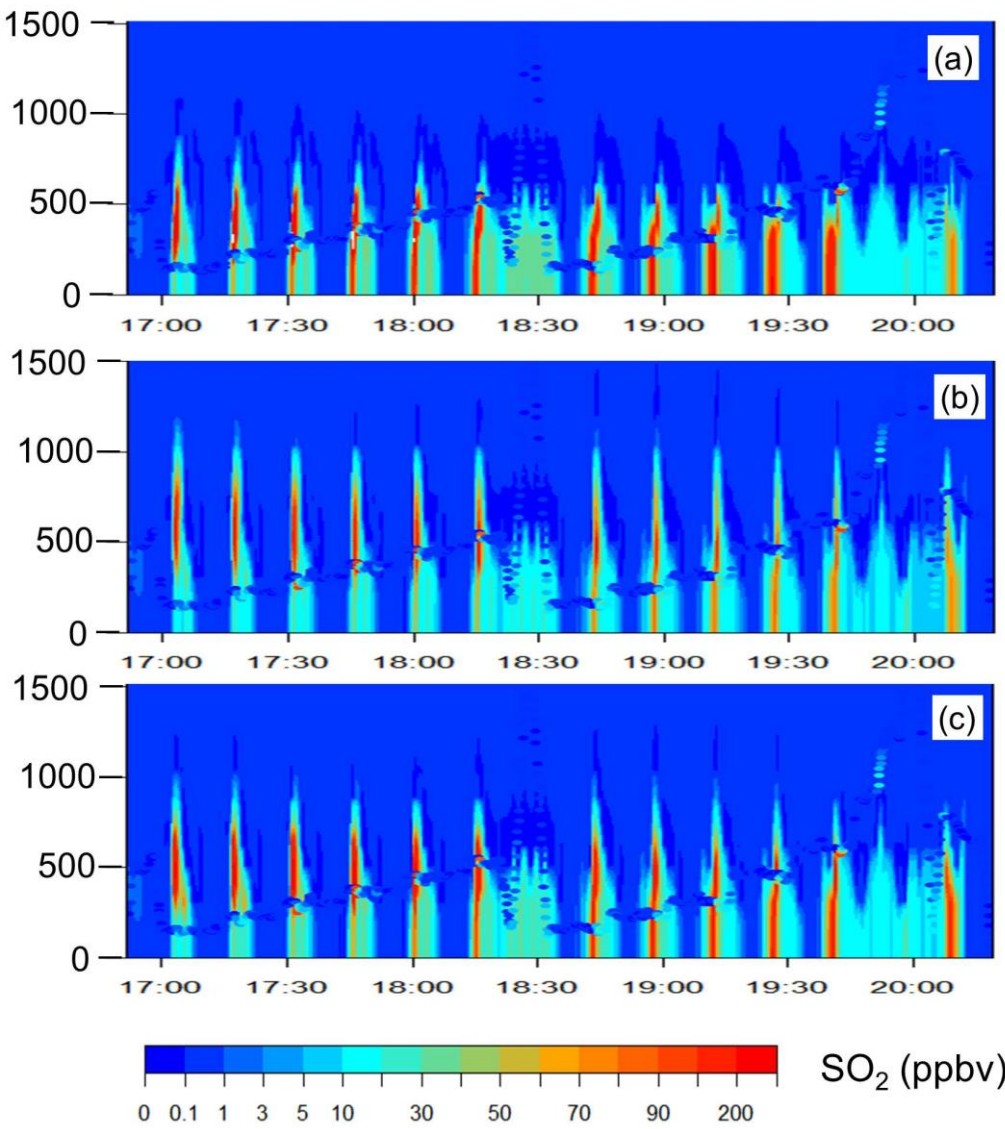

Figure 5: Model $SO_2$ profile along the aircraft path for Flight 12 (a) "12-bin" simulation (original plume-rise algorithm), (b) "Plume rise" simulation (revised plume rise algorithm), and (c) "Hourly" simulation (revised plume rise algorithm combined with hourly data for volume flow rates and stack temperatures). Panels (a-c) show model predictions in the column of the aircraft trajectory as concentration contours – aircraft observed values at the aircraft locations at the given time are shown as coloured dots overplotting the background contours.



While the aircraft values are difficult to discern in Figure 5, the collected aircraft $SO_2$ observation data at successive plume intersections during each of the two intervals were extracted from the data record, arranged so that "first plume intersection" values were vertically aligned, and the vertical intervals between these successive aircraft passes were linearly interpolated in the vertical to yield observation-based cross-sections of $SO_2$ mixing ratios, for each of the two time intervals. These are

compared to the model plumes between 17:42 and 17:54, and 19:08 and 19:25, in Figure 6(a,b), respectively. In the first interval (Figure 6(a)), the observed plume (far right profile) can be seen to be completely detached from the surface, with concentrations < 3 ppbv located below a > 100 ppbv region between 460 and 520m elevation. All three model plumes show more fumigation than the observations, with the "Plume rise" simulation showing the least fumigation of the three simulations, and the 12-bin simulation showing the most fumigation. In the second interval (Figure 6(b)), the observed

plume is located significantly higher than the model plumes (the "Plume rise" simulation plume is the closest of the three in terms of elevation, but all three model plumes underestimate the plume height by several hundred metres). While the observed plume during this second interval shows some signs of fumigation at the lowest elevation, the observed concentrations at lowest aircraft elevation are less than 30 ppbv, while the lowest model mixing ratios in the fumigation region are approximately 70 ppbv for the "Plume rise" simulation, and above 100 ppbv for the other two simulations. The

case study thus echoes the statistical analysis of Figures 3 and 4: all model simulations tend to under-predict the plume top, and overpredict the extent of fumigation, for Flight 12.





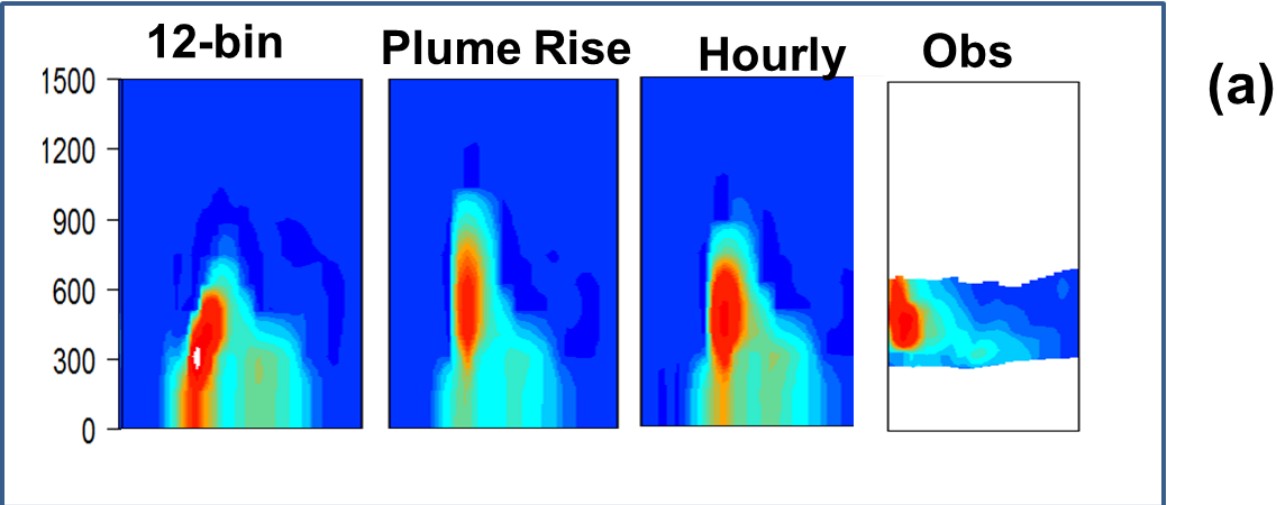

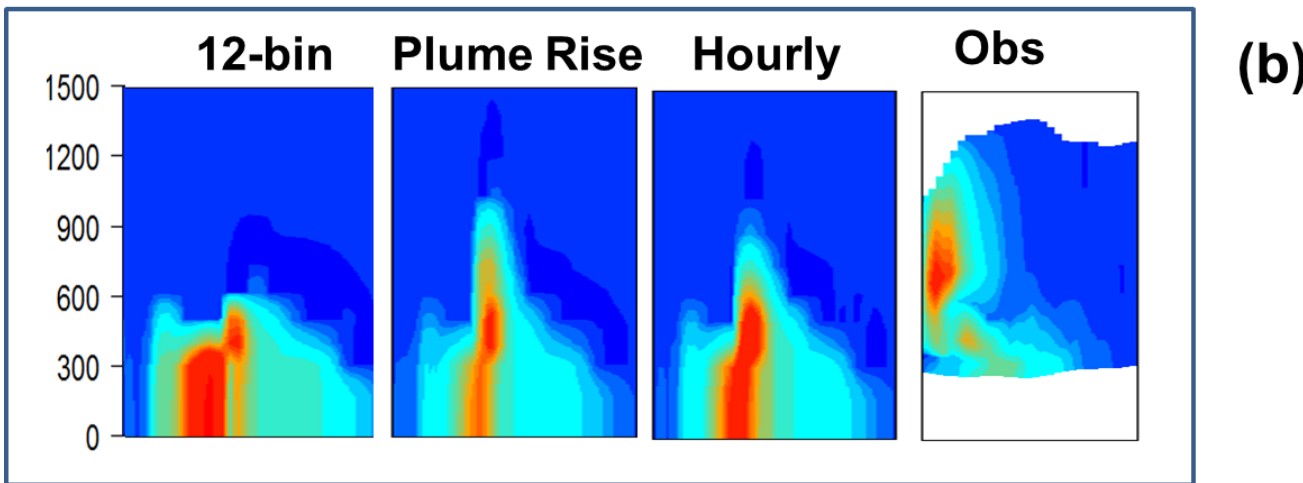

Figure 6. Zoomed-view of Figure 5. (a) 17:42-17:54, observations interpolated from successive flight passes between 17:00 and 18:19. (b) 19:08-19:20, observations interpolated from successive flight passes between 18:42 and 19:45

While the comparison is encouraging in that both of the simulations employing the new plume rise algorithm (Figure 6(b,c)) out-perform the original (Figure 6(a)) for most metrics (Figure 5(f)), the use of the CEMS-observed volume flow rates and temperatures with the new algorithm result in a degradation of performance, relative to the simulation making use of annual averages for these parameters. That is, the believed-to-be-more-realistic stack parameters result in slightly worse

10  performance: a cause for concern. The average of the hourly observed volume flow rates and temperatures for this facility's stack during flight 12 are 581.5 $m^3s^{-1}$ and 472.69K, respectively, while the corresponding annual reported values are 1174.5





m$^3$s$^{-1}$ and 513.2K. With respect to equation (1), the relative ratio of the buoyancy flux with these two sets of parameters will be:

$$R = \frac{V_r T_{s,o}(T_{s,r}-T_a)}{V_o T_{s,r}(T_{s,o}-T_a)} \qquad (10)$$

Where the subscripts $r$ and $o$ indicate the annual reported and hourly observed values of each quantity.   Assuming an

ambient temperature at stack height of 291K, the value of R is 2.28; that is, the initial buoyancy flux of the Plume rise simulation is over double that of the Hourly simulation. The hourly values are known to be more realistic during the period simulated – the revised algorithm, while providing better results than the original, thus still has a tendency to under-predict the plume heights. In our companion paper, we found that the revised algorithm (therein referred to as the "layered approach") had no significant advantages over the original Briggs algorithms – here we have found this revised approach has

considerable benefit, while showing the same overall tendency to under-predict plume heights as in our companion paper.

In order to demonstrate the extent to which the plume rise values themselves differ between flights, we have compared the calculated plume heights from each of the three algorithms examined here for 8 stacks (located at the Syncrude, Suncor, and CNRL facilities) against observations during the course of the study (Figure 7). The observed plume rise values here were derived from estimates of the SO$_2$ plume centres from the aircraft campaign's emission box flights as estimated in our

companion paper (Gordon *et al*. (2018)). Despite the differences visible in Figures 5 and 6, for flight 12, Figure 7 shows that the revised algorithm has a significant impact on calculated plume heights, greatly increasing the number falling within a factor of two of the observations (>70%), while the original algorithm has the majority of calculated plume heights falling below the 1:1 line, in accord with Gordon *et al*. (2018)). However, the impact of the differences in volume flow rates and temperatures (Figure 7(b) versus Figure 7(c))  are usually relatively minor, with the exception of a few additional points

falling below the 1:2 line for the Hourly (Figure 7(c)) simulation. The large deviation between the annual reported and measured stack parameters for flight 12 may thus be an anomaly relative to the entire record across all 8 stacks examined here. Nevertheless, Figures 3 to 6 suggest that all of model simulations have a tendency to overestimate fumigation, so we continued our examination using Flight 12 as a case study.

The model concentrations of primary pollutants are also modified by vertical diffusion and advection. The use of a plume

rise algorithm simultaneously with vertical diffusion implies the potential for "double-counting" of some proportion of the vertical mixing, in that the observation-based plume rise algorithms *de facto* incorporate vertical diffusion in their estimates of plume rise, while air-quality models must apply diffusion at all model grid-squares, including those in which plume rise algorithms have already distributed emitted mass in the vertical. If the relative impact of vertical diffusion versus buoyant plume rise is strong, this may result in excessive vertical mixing; the model effectively "double-counting" the vertical

diffusion component of the net rise. The potential for overestimates of model diffusivity magnitudes resulting in excessive vertical mixing to the ground was investigated by carrying out a sensitivity run for Flight 12 in which diffusivities in the column were halved prior to their use in calculating vertical diffusion. This sensitivity run showed a minimal impact on



model results – the magnitude alone of vertical diffusion did not influence the fumigation noted below. However, this test did not examine the potential changes associated with different magnitude changes in diffusivity as a function of height.

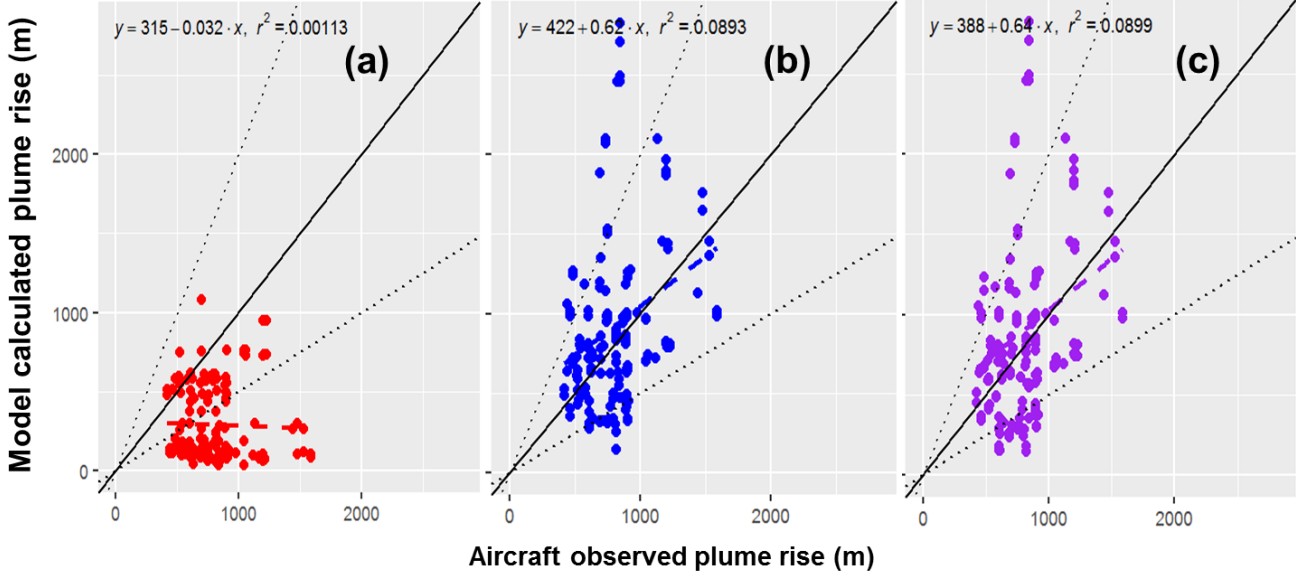

Figure 7: Observed plume rise heights during aircraft emission box flights compared to model calculated plume rise using; (a) the original plume rise algorithm; (b) new plume rise algorithm; and (c) new plume rise algorithm and CEMS hourly stack temperature and volume flow rate.

All of the plume rise algorithms are limited by the accuracy of the on-line model to accurately predict the meteorological quantities required in equations (1) through (9). We note that the original Briggs' algorithms (equations 1 through 8) are more strongly dependant on the model's ability to accurately predict meteorological conditions close to the surface, at stack height, as well as bulk parameters such as the Obukhov length, while the revised algorithm (equations 1,5 and 9) are more strongly dependent on the model's ability to accurately predict the temperature profile throughout the column.

We examined the model's temperature predictions, and compare to observations aboard the aircraft in Figure 8. Figure 8(a) shows the model-predicted temperatures in the columns around the Syncrude facility as colour contours in height versus time, similar to the mixing ratio cross-sections of Figure 5. The corresponding aircraft temperatures are over-plotted on Figure 8(a) as coloured dots employing the same temperature scale as the model values. The aircraft values, particularly in the first of the two emissions spiral periods (bracketed by vertical dashed lines in Figure 8) suggest that the model temperatures are biased high in the lowest part of the atmosphere. Figure 8(b) shows the temperature cross-sections interpolated from aircraft observations collected during the portion of the aircraft flight track crossing the $SO_2$ plume, to represent the average temperature profiles in each of the two regions. The first of these two cross-sections show an observed temperature inversion at the lowest aircraft altitudes, absent in the model temperature profile. In the second profile of Figure 8(b), the inversion is no longer apparent. The model-predicted temperatures in the lowest part of the atmosphere are also



biased high relative to both observation-based temperature cross-sections (compare Figure 8(b) corresponding to the dashed line bordered regions of Figure 8(a)). Figures 8(c) and 8(d) show the variation between model and observed temperatures in two other ways; as a pair of temperature time series during the model flight (Figure 8(c)) and as a scatterplot showing the differences in temperature (observed – model) as a function of height (Figure 8(d)).

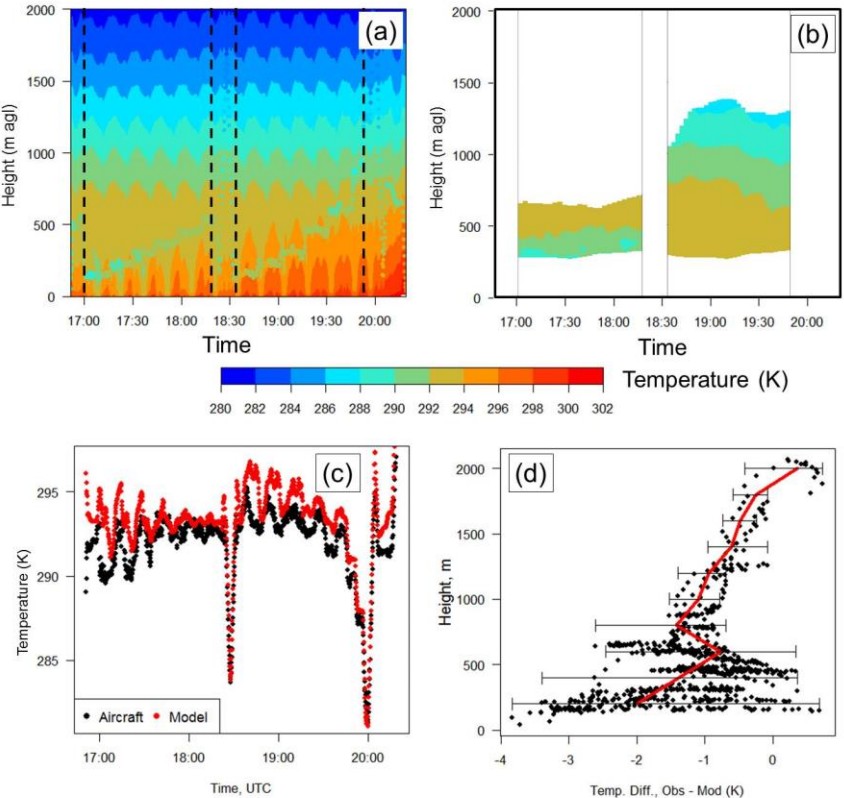

Figure 8: Model versus observed temperatures, Flight 12. (a) Background model-predicted temperature profiles with observed temperatures overlaid (dots) with the same colour scale. (b) Observed temperatures along the portion of the transect containing the plumes, between 17:00 and 18:18, and 18:32 and 19:45. (c) Model (red) and observed (black) temperatures as
10 a function of time; (d) Temperature deviation (observations – model) as a function of height, with the red line showing the mean deviation at every 100m.

All of these temperature comparisons suggest that, for Flight 12, the model tended to have positive temperature biases near the surface, biases which gradually decreased with height (Figure 8(d)). The model atmosphere would thus be expected to be less stable than the observed atmosphere, with temperature gradients reduced in magnitude relative to observations. The
15 model also reported positive values of the Obukhov length during the period (neutral to stable atmospheres; the Briggs' formula employed would be equations (3) or (4)), while the smaller magnitude temperature gradients in the model will drive parameter $s$ (equation (5)) to smaller values. While $s$ features in the stable atmosphere formula (3), it does not feature in the



neutral atmosphere formula. That is, the original Briggs' formulae are relatively insensitive to errors in the temperature profile in near-neutral conditions, with only a weak influence via the $F_b$ term. However, the revised algorithms (equations (9), (1), and (5)) will be influenced by the accuracy of the temperature gradient at every point throughout the temperature profile. This analysis suggests that the original Briggs' algorithms (the "12-bin" simulations) will be less influenced by the temperature errors shown in Figure 8(d), while the revised approach ("Plume rise" and "Hourly" will be more influenced by them, contributing to higher estimates of plume heights.

This particular case study thus places an important caveat on our results – while the revised plume rise approach provides better results, and a better estimate of plume rise relative to the observations, it may be doing so in part in response to a model overestimate of surface heating and the corresponding reduction in the magnitude of the temperature gradient, to which the latter algorithm is sensitive, and to which the original Briggs' algorithms are less sensitive.

Our final analysis examines the effects of the different plume rise algorithms on the broader region, through comparisons of multi-week average differences of surface and downwind vertical cross-section mixing ratios of $SO_2$ (Figure 9). The change in $SO_2$ ("Plume Rise" – "12-bin") average surface mixing ratio and a representative cross-section are shown in Figure 9(a,b), while the corresponding differences for the two simulations employing the revised algorithm ("Hourly" – "Plume Rise") are shown in Figure 9(c,d). The first comparison (Figure 9(a)) shows the substantial impact of the revised plume rise algorithm relative to the original Briggs' formulation; surface concentrations of $SO_2$ have decreased over most of the domain, often by by up to tens of percent. The corresponding cross-section (Figure 9(b)) shows that most of the $SO_2$ removed from the surface is transported aloft, resulting in substantial relative increases in $SO_2$ mixing ratios throughout the lower troposphere. The second comparison (Figure 9(c,d)) shows that the use of hourly CEMS stack parameter data results in substantial local increases and decreases – changes in plume height associated with the use of the hourly stack parameters are sometimes responsible for both positive and negative changes in the tens of percent, relative to the simulation driven by annual reported stack parameters. The $SO_2$ mass formerly being carried aloft now fumigates downwind, in the "Hourly" cross-section.

In similar evaluations for $NO_2$ (not shown), percentage differences of up to 10% in $NO_2$ surface mixing ratio and less than 1% maximum difference is surface ozone mixing ratio for the 30-day average period were found. The choice of a plume rise algorithm thus has a substantial impact on average surface and lower troposphere concentrations of those species predominantly emitted from large stacks.



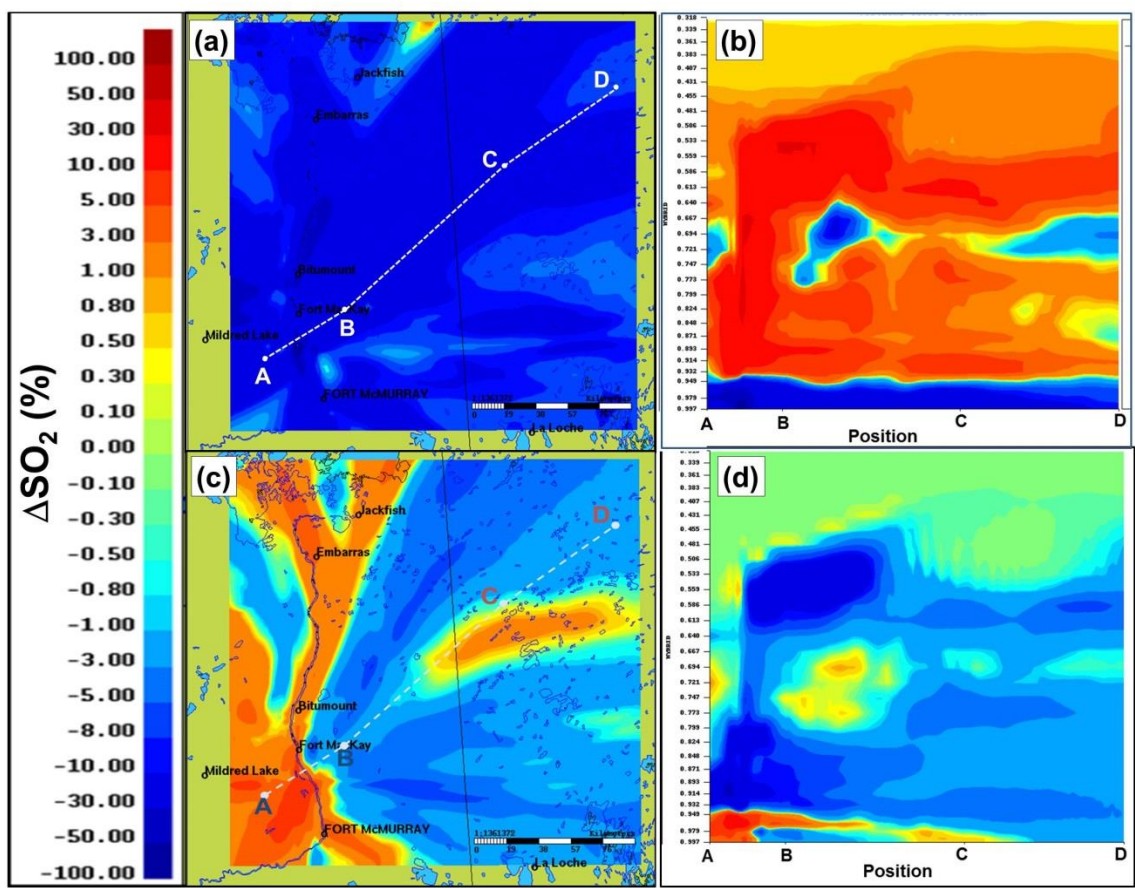

Figure 9. Comparison of model-generated average mixing ratios: percent differences in multi-week averages. (a) Average surface mixing ratio percent differences for "Plume rise" – "12-bin". (b) Average cross-section percent differences along cross-section A→B→C→D, for "Plume rise" – "12-bin" . (c) Average surface mixing ratio percent differences for "Hourly" – "Plume rise". (d) Average cross-section percent differences along cross-section A→B→C→D, for "Hourly" – "Plume rise".

## 5 Conclusions

We have carried out a set of four model scenarios for a 2.5-km resolution nested domain using the GEM-MACH air quality forecast model for the Athabasca oil sands region of Alberta, Canada. These scenarios have allowed us to examine the relative impacts of aerosol size distribution and plume rise algorithms on model performance, relative to surface and aircraft observations of multiple chemical species.

While a 2-bin configuration with sub-binning of microphysical processes has been employed in the past for operational forecasting due to computational processing time constraints (Moran *et al.*, 2010), we find that the 12-bin configuration has better performance for all surface $PM_{2.5}$ prediction metrics, including an overall 34% reduction in the magnitude of the bias of $PM_{2.5}$, for a 25% increase in processing time.



Comparisons with the model and observed stack plumes showed that all algorithms tended to under-predict plume heights, in accord with our companion measurement-driven investigation of plume rise using the Briggs (1984) plume rise algorithm (Gordon et al., 2018). However, in contrast to that work, significant improvements to model performance were found with the adoption of a revised plume rise algorithm, also based on Briggs (1984), in which local changes in stability in individual

model stable and neutral model layers are used to calculate the fractional reduction in buoyancy of the rising plume. Tests of the revised algorithm using both annually reported stack parameters and hourly parameters from Continuous Emissions Monitoring both resulted in significant improvements in model performance in comparison to the original approach. However, the use of hourly observed (and presumed more accurate) stack parameters resulted in degradation of performance relative to the use of annual reported values for these parameters. Further investigation using a specific case study suggested

that the improvements associated with the revised algorithm may in part be due to model positive biases in lower atmospheric temperature, resulting in model underestimates in the magnitude of atmospheric temperature gradients. Nevertheless, the revised approach was found to correct much of the predominantly negative bias in predicted plume height seen for Briggs' original algorithms, correcting the biases in plume height noted in our companion paper, in which the algorithms were driven using observed meteorology.

Despite these improvements, and the tendency of the model to underestimate temperature gradients, the model still over-predicts the extent of fumigation for all plume rise algorithms tested, implying the need for further work. The revised approach found to be the most favorable in the current work is based on two key parameters; entrainment coefficients determined by Briggs from data collected in 1975 to be approximately 0.08 and 0.4 respectively; we recommend that these coefficients be re-estimated using more recent data.

Our simulations have shown that the choice of a plume rise parameterization has a very significant impact on downwind concentrations of $SO_2$ from the oil sands sources, with the approaches having the more accurate plume heights also resulting in significant reductions in surface $SO_2$, and increases in $SO_2$ aloft, helping to correct pre-existing positive and negative biases in the model at these elevations. Smaller impacts were found for $NO_2$, and minimal impacts for ozone.

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
