# Peer review of "A chemical transport model study of plume rise and particle size distribution for the Athabasca oil sands"

_Atmospheric Chemistry and Physics, 2018_

## Referee Comment (RC1) · Anonymous Referee #1 · 26 Mar 2018

The manuscript by Akingunola et al. presents an interesting set of sensitivity simulations to plume rise modelling from stacks in the framework of the Canadian mesoscale chemistry-transport eulerian model GEM-MACH. The study is timely since simulation of the subgrid plume dynamic processes are still affected by significant uncertainties, as also confirmed by this work, and it is relevant for air quality applications considering the large role that emissions from elevated stacks plays nowadays and will play also in the near future. I found the manuscript generally well written and clear and I recommend publication to ACP after some minor corrections and clarifications as detailed below.

- P. 1, L. 19: "...reducing the magnitude of the original surface PM2.5 negative biases by 32%". Would be more clear to specify the range of change: from bias x to bias y.

- P. 1, L. 24: "...with 39 to 60% of predicted plume heights ...". I suggest to specify what the given range is referring to, e.g. rephrasing the sentence adding a compact summary of what are the best and worst performing cases.

- P. 1, L. 28: "...between the surface and 1km elevation". I suggest to clarify the concept. From my understanding this refers to the bias in the simulated lapse rate dT/dz as compared to observations.

- P. 2, L. 6: "...(it is not created by chemistry)". Suggest to change "chemistry" in "photochemical reactions in the atmosphere".

- P. 2, L. 10-11: "Anthropogenic SO2 emissions are the main source of most atmospheric sulphur deposition". Suggest to add a reference for this statement.

- P. 3, L. 17-18: Please specify to what conditions/cases the given ranges (34 to 52% and 0 to 11%) are referred to.

- P. 3, L. 22: typo "Sulpher" should be "Sulphur"

- P. 4, L. 15: typo "as-phase" should be "gas-phase"

- P. 5, L. 2: Would be more informative to add the height of the levels in the bottom 1 km of the model.

- P. 5: Moreover, given the relevance for the results, I recommend to add a description of the parameterizations adopted in the model for the PBL and the surface layer turbulence.

- P. 8, eq. 6: Please check the second condition "0.5 < H < 1.5" since the range seems to refer to a unitless quantity, but here only H is given.

- P. 9, L. 6: "top of the atmosphere" is confusing: is it perhaps the top of the PBL?

- P. 9, L. 8: "value of hs was assumed", perhaps is "value AT hs was assumed". Moreover, the "s" of "hs" should be a subscript.

- P. 10, L. 5: typo "and = hs", please check the left-hand side.

- P. 11, L. 21: "modstat" should be "modStat"

- P. 12, L. 27: "...negative bias has decreased by 34%" it is not perfectly clear here and in the following if these bias changes are actual relative changes or absolute changes of the normalized mean bias. Please clarify.

- P. 12, L. 31-32: "Figure 2 shows that ... less than 5 um diameter ...". Please double check this statement. The figure shows the PM2.5 concentrations binned as a function of CONCENTRATION not SIZE.

- P. 15, Table 2: Please check the values that should be given in Italics, since not all the rows seem to contain it.

- P. 16: referred to the discussion of SO2 overestimation and SO4 underestimation: can the two things be linked? E.g. by slow SO2 to SO4 conversion in the model, perhaps by slow aqueous chemistry?

- P. 17, L. 4-7: the paragraph seems to imply the presence of at least a (b) point, but only (a) is given. Please check or rephrase.

- P. 19, L. 11: "...took place between 16:30 and 20:30 on Aug 24th...". Although I am assuming the intervals are given in local time and not in UTC, it would be useful to have a confirmation in the paper. Here and also at least in the caption of the first figure showing time series (Figure 5).

---

## Referee Comment (RC2) · Anonymous Referee #2 · 17 Apr 2018

This discussion paper presents results of simulations with the operational Canadian chemistry-transport model GEM-MACH. A range of different model and input options is tested and compared with surface and aircraft measurements in the Alberta oil sands region. These options consist of different aerosol size resolution, different plume-rise formulations, and different data for the stack emission parameters. The authors found that using twelve instead of two aerosol size bins reduces the model bias for PM significantly. For the plume rise formulation, the results are less clear and the authors question whether improvements come for the right reasons. Real hourly stack data did not improve the results, and the question why remains unanswered.

This paper introduces a very interesting and potentially highly useful field campaign. It also provides important insights into the performance of the operational Canadian model. However, the paper has some weaknesses listed below.

1. In the same special issue for which this paper is submitted, there is another paper by the same group of authors, Gordon et al. 2018, which is also devoted to the plume rise topic, and it is said to have found opposite results. Neither is the reason for that clearly resolved, nor does it become clear why the plume rise topic is split between two papers.

2. The model performance is not only influenced by the aspects forming the focus of this paper, but also by the accuracy of the meteorological part of the model, and by the numerics of transport, notably the vertical diffusion and the handling of the point sources in the Eulerian framework. Their role is discussed only at the very end and, in my opinion, not sufficently in depth. In order to evaluate specific model aspects, one first needs to understand the performance of the model in general, with its strengths and weaknesses.

3. The statistical approach chosen for the evaluation of the model options relies on metrics which exclusively are based on "match in time and space" data pairs. It is well known in air-pollution modelling that for near-source conditions (which is what we find here), there is often too much "noise" in the data (be it due to the stochastic nature of the plume, be it due to unresolved meteorological variability) to give meaningful results. Correspondingly, some of the statistical parameters are not very good. Therefore, global comparisons (such as deviations from the cumulative frequency distribution, statistics of cross-wind integrated values, or average dependency on key parameters such as stability and wind speed) are often used to assess models in a more robust way.

4. The paper is written well on the "small scale" (apart from numerous technical deficiences as listed below), but the broad topics could be worked out more clearly.

In the end, the findings are: twelve aerosol size bins are better than two (not surprising, but good to see it quantified), there is an improvement by using the model's vertical profile information for plume rise calculation but given the model's deficiencies the overall conclusion seems to be not so clear, and no improvement was found for using hourly stack data, but it remains unresolved why. We may wonder whether the work is mature enough for publication if we consider this state of the quintessential findings.

**Specific comments**

1. Page 2, l. 18: Why are you thinking that reasons for weak performance include only those meteorological variables that are used for the plume rise calculation, but not, for example, wind direction?

2. The model overview section lacks information on the numerical scheme used for vertical diffusion even though this is crucial in the context of study (cf. discussion on p. 24). The main reference for the MACH model seems to be Makar et al. (2010) – an extended abstract that would not be available for most people who haven't attended the conference as it is not freely accessible. Is there no more detailed and open description of this model? Note that also the Coté et al. citation is one of those for which the reference is missing. In addition, the handling of the point sources is not described (usually, Eulerian models use some sub-model to track plumes until they match the size of the grid cells).

3. The model set-up description in section 2.2 is not easy to follow. It might be helpful to move some of the information into a table and to shorten the text.

4. Page 8, line 1: The plume's buoyancy flux is **not** dependent on the stack height (at least not directly).

5. From the sentence beginning on p. 11, l. 7, on, the text does not really belong to the section 2.2.3. It should become a section of its own, as it introduces the simulations forming the base of the rest of the paper (maybe merge with some parts of the 2.2 chapeau).

6. Page 11, Section 3.1: What is the justification for removing measurements with values exceeding some threshold? Without proper justification this would not be acceptable.

7. Page 12, Section 3.2: The phrase 'spatial linearly interpolated model values at the models chemistry time resolution of 2 minutes' is awkward. If you have 10 s data as said before, why do you need to interpolate for obtaining 2 min data? Also, it would be good to know which distance corresponds to both 10 s and 2 min flight data, and how this compares to the model's grid size.

8. Section 4 (Results and Discussion) needs to be structured into subsections.

9. Table 1: Apart from widely used or self-explanatory metrics such as FAC2, RMSE or *r*, the metrics parameters need to be defined.

10. Page 12, l. 31: "Figure 2 shows that the model simulations are biased high for particles less than 5 $\mu$m diameter, and biased low for the larger particle sizes." As this figure only shows results for PM2.5, a statement on larger particles can't be based on it.

11. Page 13, l. 14: Information on the bin sizes belongs to the model description section, not the result section.

12. Page 13: The second paragraph on this page contains a number of statements about results without pointing to the figures or tables which show them.

13. Concerning the model performance for PM, it should be discussed that even though the twelve-bin version leads to significant improvements, major discrepancies to observations remain.

14. A number of tables are presented where several metrics are used to compare various model versions, with the best one being emphasised by bold print. Sometimes, differences are tiny and probably insignificant. Only those values that are *significantly* better should be highlighted to avoid a wrong impression of the results (for example, in Table 3 the model version seems to have no impact for $O_3$ but we get the impression that the simpler model is better.)

15. Why is the use of hourly emission data beneficial for $NO_2$ but detrimental for $SO_2$?

16. The discussion paper does not comply with the ACP Data Policy; it does not have a "Data availability" section and says nothing about data availability.

**Technical comments**

In this manuscript, there are many details that need to be corrected or improved. I hope that I have identified most of them, nevertheless I would call upon the authors to generally pay more attention to those details in their revised manuscript.

The following list mentions such topics either globally or, for some, individually.

1. Page 2, l. 14: CEMS has nothing to do with observations.

2. Spelling mistakes that are not caught by a spell checker. I noticed for example "RAND" instead of "RANS" on page 3, line 11, but there may be more. Please proofread your manuscript.

3. Page 3, line 11ff.: This sentence is a bit awkward, rephrase it.

4. Writing of numbers with units: don't forget to leave a (at least thin) space between them.

5. Punctuation: There are instances of doubled or missing punctuation, or inappropriate use of the hyphen.

6. Write out numbers from one to twelve if they occur in the running text.

7. Page 3, l. 22, "lower values" would fit better here than "lower elevations" (elevation refers to topography).

8. Page 4, l. 8: Create a proper reference for SMOKE instead of putting it into the text.

9. Page 5, l. 5: "data-assimilated meteorological analyses" is not proper English.

10. Fig 1: Do not confuse the indication of subfigures with the indication of domains in the first subfigure. Due to the dark background, subfigures d) and e) are not readable. It might be better to show topography or land-use in a simple way rather than a satellite image. Why do they show two different domains? Also, I don't understand what "Google-Earh-referenced" means.

11. On page 8, please pay attention to how you incorporate the equations into the surrounding text (what is a full sentence?)

12. Case distinction for Eqs. 2, 3, 4: While the conditions are correctly formulated, they might be easier to grasp if they were expressed in terms of $h_s$ instead of $L$, adding a second condition (with the sign of $L$).

13. Page 8: I have not compared these (or other) equations with the original source (Briggs), but given the number of typos in this manuscript I would recommend that someone should do this.

14. Eq. 5: do not use a differential for a finite difference (write with $\Delta$ instead).

15. Eq. 6, middle case: $H$ has to be replaced by the fraction used in the other cases.

16. Eq. 8: It would be more easy to simply write $h_t$ and $h_b$ – full-word subscripts are not standard in math notation and a bit cumbersome.

17. Page 9, line 6: I don't think that you really mean *top of the atmosphere* here. Also, a few lines down, hs should become $h_s$.

18. Page 10: Do not write text in Eq. 9 in math mode, but add math mode for the eqations embedded in the following text.

19. Put "Eq." in front of the reference to an equation.

20. Page 11, l. 21, quote 'openair'.

21. Spelling of PM is inconsistent, sometimes PM2.5, sometimes $PM_{2.5}$, similar for PM1. For simplicity, I suggest not to use subscripts here.

22. Page 13, l. 13, $\mu$ is missing.

23. Page 13, l. 16: "compared similarly" is not proper English. Line 20, not flights are "cloud-free" but the atmosphere probed. Line 20, "particle ammonium", probably "particulate ammonium".

24. Figures: Figures should be of uniform appearance and layout, and of professional quality. Some aspects that should be improved include

   - Do not frame the figures.
   - Axes, plot frames etc. should be in black, not in colour or light gray.
   - Do not use boldface in figures and use a sensible font size.

   - Move legend frames into corner of plot.
   - If there are subfigures, place a), b) etc. above or to the left of the subfigure, not within the plot.
   - Figure 3: The three subfigures show identical results, with the only difference that b) and c) zoom in on a part of the bins. This is not needed.
   - Figure 4: Again, zoomed versions are not needed.
   - Figure 5: The quality of the figures in insufficient; one can not see well the dots of the aircraft. I assume that those dots would be in the color corresponding to the airborne measurement even though this is not said explicitly. A part of the problem might be that dots are too big and overlap. A solution might be not to show many passes through the plume, but only a few characteristic ones with a better temporal resolution. I include here a zoomed-in detail of the plot to illustrate the quality problem.

[Figure]

   - Fig. 7: The background should not be white on light gray, but black on white. Also, make each plot square.
     The caption says that *plume-rise heights* are depicted, while the corresponding text (p. 24, l. 13) says that *plume heights* were calculated. As the physical stack height is not zero, this is not the same.
   - Fig. 9: The map scale is not readable in a) and c), while in b) and d) the

y-axis annotation is too tiny. Also, a geographical grid or coordinate axes are desirable for the map plots.

- Fig. 12a: Similar to to Fig. 5, the dots representing the aircraft measurements are not properly visible.

25. Tables: I hope that the table layout will be taken care of by the technical editor of ACP. Column headings should not be be broken within a word, and the use of italics and boldface has to be restricted to where it actually indicates something, not for the names of parameters etc.

26. Table 1: The full names of the metrics can go into the caption and the table itself would use only abbreviations.

27. Table 3: The caption should refer to the place where the abbreviations are defined.
The caption speaks about "concentration measurements" while the table reports ppb. Also, it is not stated whether mixing ratios are volume or mass mixing ratios.

28. Do not number the References section.

29. The following reference entries seem not to be mentioned in the paper:
EPA (no year) State-Level SO2 Data, Liggio et al. 2017, Vet et al. 2014. Note also that if there is no year, a text such as "no year" is usually substituted for it.

30. The following quotations in the text lack a matching entry in the References section:

- Briggs (1985) quoted on p. 9, l. 22, is missing.
- Coté et al. (1998), quoted p. 4., l. 6, is missing.
- Im et al. (2015) quoted on p. 3, l. 2, lacks the a/b indicator.

- JOSM (2011) is not labelled JOSM in the References.
- Stroud (2008) quoted on p. 4, l. 13, should be Stroud et al. (2008).
- Stroud (2016) quoted on p. 4, l. 11, is missing.

31. There are several instances of different spelling of authors' names in the body of the manuscript and in the Reference section, e.g. Giebel vs. Gielbel, Hanna vs. Hann

32. The formatting of journal names is inconsistent.

33. The abbreviations used for journals are very short ones, not the standard, for example *Atm. Env.* instead of *Atmos. Environ.*, or even totally non-standard such as *J. of App. Met.*. The references to the ACP Special Issue on Oil Sands of which this manuscript is a part should be uniform.

34. EPA 2018 a and b lack a title. Also, if EPA 2018b is from 2015 as the URL suggests, why do you label it EPA 2018?

---

## Author Response (AR1)

**Response to Reviewers**
**A chemical transport model study of plume rise and particle size distribution for the Athabasca oil sands**

Original Reviewer comments are in normal font, responses in *italics*.

**Anonymous Referee # 1**

The manuscript by Akingunola et al. presents an interesting set of sensitivity simulations to plume rise modelling from stacks in the framework of the Canadian mesoscale chemistry-transport eulerian model GEM-MACH. The study is timely since simulation of the subgrid plume dynamic processes are still affected by significant uncertainties, as also confirmed by this work, and it is relevant for air quality applications considering the large role that emissions from elevated stacks plays nowadays and will play also in the near future. I found the manuscript generally well written and clear and I recommend publication to ACP after some minor corrections and clarifications as detailed below.

*We thank the reviewer for the helpful comments; our responses to the minor corrections and clarifications follows.*

1. P. 1, L. 19: ". . .reducing the magnitude of the original surface PM2.5 negative biases by 32%". Would be more clear to specify the range of change: from bias x to bias y.

*The text has been modified to read "...reducing the magnitude of the original surface PM2.5 negative biases 32%, from -2.62 to -1.72 $\mu g \ m^{-3}$."*

2. P. 1, L. 24: ". . .with 39 to 60% of predicted plume heights . . .". I suggest to specify what the given range is referring to, e.g. rephrasing the sentence adding a compact summary of what are the best and worst performing cases.

*Our submitted draft referred to the companion paper's results in the abstract – rather than take examples from that work, we've carried out additional analysis from our scatterplot figure (Figure 8 in the revised manuscript), and based on that analysis, we have modified the sentence in the abstract to read:*
*"As in our companion paper (Gordon et al., 2018), we found that Briggs algorithms based on estimates of atmospheric stability at the stack height resulted in under-predictions of plume rise, with 116 out of 176 test cases falling below the model:observation 1:2 line, 59 cases falling within a factor of 2 of the observed plume heights, and an average model plume height of 289 m compared to an average observed plume height of 822 m. We used a high resolution meteorological model to confirm the presence of significant horizontal heterogeneity in the local meteorological conditions driving plume rise. Using these simulated meteorological conditions at the stack locations, we found that a layered buoyancy approach for estimating plume rise in stable to neutral atmospheres, coupled with the assumption of free rise in convectively unstable atmospheres, resulted in much better model performance relative to observations (124 out of 176 cases falling within a factor of two of the observed plume height, with 69 of these cases above and 55 of these cases below the 1:1 line and within a factor of two of observed values). This is in contrast to our companion paper, wherein this layered approach (driven by meteorological*

*observations not co-located with the stacks) showed a relatively modest impact on predicted plume heights. "*

*We have also included a new table (Table 7 in the revised manuscript) which includes the distribution of values about the 1:2, 1:1 and 2:1 lines of the scatterplots, as well as the predicted average plume heights, in order for this comparison to be more quantitative.*

3. P. 1, L. 28: ". . .between the surface and 1km elevation". I suggest to clarify the concept. From my understanding this refers to the bias in the simulated lapse rate dT/dz as compared to observations.

*We have changed the sentence to read, "Persistent issues with over-fumigation of plumes in the model were linked to a more rapid decrease in simulated temperature with increasing height than was observed. This in turn may have led to overestimates of near-surface diffusivity, resulting in excessive fumigation."*

4. P. 2, L. 6: ". . .(it is not created by chemistry)". Suggest to change "chemistry" in "photochemical reactions in the atmosphere".

*We've followed your suggestion; the sentence now reads: "SO$_2$ is a primary emitted pollutant (it is not created by photochemical reactions in the atmosphere), with the majority of anthropogenic SO$_2$ emissions in the study region coming from large smokestacks (Zhang et al., 2018)."*

5. P. 2, L. 10-11: "Anthropogenic SO2 emissions are the main source of most atmospheric sulphur deposition". Suggest to add a reference for this statement.

*We have added a reference to Mylona, 1996 in the updated manuscript to support the above statement.*

6. P. 3, L. 17-18: Please specify to what conditions/cases the given ranges (34 to 52% and 0 to 11%) are referred to.

*The sentence refers to the companion paper, which did not segregate the extent of over/underprediction by stability class; we have added to the original statement, viz: "There we found that the Briggs (1984) plume rise parameterization significantly underpredicted plume heights in the vicinity of the multiple large SO$_2$ emissions sources in the Canadian Athabasca oil sands, with 34 to 52% of the parameterized heights falling below half of the observed height, compared to 0 to 11% of predicted plume heights being above twice the observed height, over conditions ranging from neutral, through stable to unstable."*

7. P. 3, L. 22: typo "Sulpher" should be "Sulphur"
*Corrected.*

8. P. 4, L. 15: typo "as-phase" should be "gas-phase"
*Corrected.*

9. P. 5, L. 2: Would be more informative to add the height of the levels in the bottom 1

km of the model.

*The model uses a hybrid coordinate system so the heights above ground change with location. However, the revised manuscript includes a new Figure (Figure 2 in the revised manuscript) which shows the model heights at a number of locations in the portion of the 2.5km domain studied here.*

10. P. 5: Moreover, given the relevance for the results, I recommend to add a description of the parameterizations adopted in the model for the PBL and the surface layer turbulence.

*The revised manuscript contains a new table (Table 1) which includes the main parameterizations in the meteorological model, including the Moist TKE scheme used for turbulence.*

11. P. 8, eq. 6: Please check the second condition "0.5 < H < 1.5" since the range seems to refer to a unitless quantity, but here only H is given.

*This was a typo on our part, and has been corrected in the revised manuscript.*

12. P. 9, L. 6: "top of the atmosphere" is confusing: is it perhaps the top of the PBL?

*An error on our part – the portion of the sentence should have read (and has been corrected to) "between $h_t$ and $h_b$".*

13. P. 9, L. 8: "value of hs was assumed", perhaps is "value AT hs was assumed". Moreover, the "s" of "hs" should be a subscript.

*The phrase was modified to read "centered on $h_s$"*

14. P. 10, L. 5: typo "and = hs", please check the left-hand side.

*The sentence has been modified to "At the stack height, $F_{j=0} = F_b$, and $z_j = 0$ (that is, the vertical distances are relative to the top of the stack)."*

15. P. 11, L. 21: "modstat" should be "modStat"

*Done.*

16. P. 12, L. 27: ". . .negative bias has decreased by 34%" it is not perfectly clear here and in the following if these bias changes are actual relative changes or absolute changes of the normalized mean bias. Please clarify.

*The sentence has been changed to ". For example, the magnitude of the mean bias has decreased from -2.623 to -1.725 µg m$^{-3}$, a reduction of 34%, indicating that a sizeable fraction of particulate under-predictions in 2-bin simulations may be due to poor representation of particle*

*microphysics through the use of the 2-bin distribution, despite sub-binning being used in some microphysics processes. "*

17. P. 12, L. 31-32: "Figure 2 shows that . . . less than 5 um diameter . . .". Please double check this statement. The figure shows the PM2.5 concentrations binned as a function of CONCENTRATION not SIZE.

*Thanks for catching this – the reviewer is quite right; this has been corrected.*

18. P. 15, Table 2: Please check the values that should be given in Italics, since not all the rows seem to contain it.

*Corrected.*

19. P. 16: referred to the discussion of SO2 overestimation and SO4 underestimation: can the two things be linked? E.g. by slow SO2 to SO4 conversion in the model, perhaps by slow aqueous chemistry?

*The aircraft observations were conducted under clear-sky conditions, so the potential for aqueous chemistry being the main issue is unlikely. Rather, noting that both predicted SO2 and SO4 aloft had negative biases, while predicted SO2 at the surface was biased high, it seems more likely that the main cause of the SO2 and SO4 negative biases aloft was a tendency for the model to overpredict fumigation to the surface, as noted in the original and revised manuscript.*

20. P. 17, L. 4-7: the paragraph seems to imply the presence of at least a (b) point, but only (a) is given. Please check or rephrase.

*The (a) has been removed (holdover from an earlier version of the manuscript, missed on checking prior to submission).*

21. P. 19, L. 11: ". . .took place between 16:30 and 20:30 on Aug 24th. . .". Although I am assuming the intervals are given in local time and not in UTC, it would be useful to have a confirmation in the paper. Here and also at least in the caption of the first figure showing time series (Figure 5).

*Corrected. The first pair of times are actually UTC (local times have been added in the revised manuscript), and Figure 5 (now Figure 6 in the revised manuscript) were also in UTC; this has been mentioned explicitly in the revised figure caption.*

**Anonymous Referee # 2:**

This paper introduces a very interesting and potentially highly useful field campaign. It also provides some important insights into the performance of the operational Canadian model. However, the paper has some weaknesses listed below.

*We thank the reviewer for the detailed comments on the paper – in addressing these, we feel that these comments have resulted in a significantly improved manuscript. The reviewer's comments and our responses (in italics) follow:*

1. In the same special issue for which this paper is submitted, there is another paper by the same group of authors, Gordon et al. 2018, which is also devoted to the plume rise topic, and it is said to have found opposite results. Neither is the reason for that clearly resolved, nor does it become clear why the plume rise topic is split between two papers.

*In our observation companion paper (Gordon et al, 2018) we saw that the Briggs algorithms, including the layered approach, tended to significantly underestimate plume rise. However, in that work we also noted that the observations themselves showed significant horizontal heterogeneity in the meteorological data used to drive the plume rise equations, with the corollary that the conditions at the actual location of the stacks may be sufficiently different from the surrounding meteorological towers to influence the predicted heights. No observations are available at the stack locations themselves – we therefore investigate this potential local influence further, using the high resolution on-line chemical transport model, GEM-MACH. As a demonstration of the model's ability to capture the local heterogeneity, we show in Figure 2(a) of the revised manuscript a snapshot of the PBL height at Saturday August 24, 2013 at 1 pm local time, and Figure 2(b) the corresponding model generated temperature profiles at the local meteorological towers AMS03, AMS05, the windRASS instrument and at three of the stacks examined in the companion paper. The model shows a significant variation in the temperature profiles between the tower and windRASS locations where the observations are available, and the stack locations. The temperature profiles suggest strong differences in both the strength of the inversion and its vertical location. This confirms the potential for spatial variability to have a significant influence on predicted plume heights relative to the meteorological observation locations in our companion paper. We therefore investigated the plume rise algorithms again within the current work, in order to determine the extent to which this local variability may influence predicted plume heights. In contrast to the companion paper, we found that the "layered" approach of calculating local stability residuals through successive model layers resulted in significantly improved plume heights relative to the more standard Briggs approaches which employ stability estimates at the top of the stacks.*

*We have included the new Figure 2 in the revised manuscript, as well as some discussion in the Introduction section of the manuscript:*

*"...Our companion paper made use of different sources of meteorological observations to drive the Briggs (1984) plume rise algorithms, as well as CEMS data and aircraft observations of $SO_2$ plumes from multiple sources over a 29-day period. There we found that the Briggs (1984) plume rise parameterization significantly underpredicted plume heights in the vicinity of the multiple large $SO_2$ emissions sources in the Canadian Athabasca oil sands, with 34 to 52% of the parameterized heights falling below half of the observed height, compared to 0 to 11% of predicted plume heights being above twice the observed height, over conditions ranging from neutral, through stable to unstable.*

*However, in our companion paper we also noted the presence of considerable spatial heterogeneity in the meteorological observations used for the algorithm tests. Temperature profiles and other data used to define the input parameters for the Briggs algorithms were taken from two tall meteorological towers, a windRASS, and a research aircraft, and showed a substantial variation in the resulting plume height predictions, despite relatively close physical proximity of these sources of meteorological data (e.g. 8 km distance between the two meteorological towers). The region under study is subject to complex meteorological conditions due to the nature of the terrain (a river valley with up to 800 m of vertical relief, and open pit mines and settling ponds which may each be tens of $km^2$ in spatial extent). This heterogeneity cast some uncertainty on the results of the companion paper, in that the best application of the plume rise algorithms would be driven by the meteorology at the location of the stacks, rather than the location of the available meteorological instruments, and the latter suggested substantial local changes in meteorological conditions. As we show in the sections which follow, the spatial heterogeneity of meteorological conditions has a controlling factor on the predicted plume rise, and, in contrast to our companion paper, an approach making use of local temperature gradients between individual model layers has greatly improved accuracy in comparison to those inferring atmospheric stability conditions from the conditions at the top of the emitting stacks."*

*The new Figure 2 is described via the following discussion in the revised text:*

*"We noted in our companion paper (Gordon et al., 2018) that meteorological observations varied substantially in the study region depending on location, citing this as a possible confounding factor on the results of tests of the plume rise algorithms. This spatial heterogeneity was well captured by the high resolution GEM-MACH simulations, as is demonstrated by the example depicted in Figure 2, which shows the typical local variation in planetary boundary layer height (Figure 2(a)), ranging from about 1200m to 400m, the lower values corresponding to the main cleared areas (open pit mines, settling ponds) of the industrial facilities. The corresponding temperature profiles in several locations marked in Figure 2(a) are given in Figure 2(b): These show a substantial difference in model predicted stability at the three meteorological observation locations of Gordon et al. (2018) (windRASS, AMS03, and AMS05), and substantial differences between these and the locations of the main stacks of some of the facilities (Syncrude 1, CNRL, and Suncor). The temperature profiles show that the height and strength of the inversion may vary by over 100m in the vertical, and that the profiles do not merge with the larger scale flow until an elevation of 750m asl (450m agl) is reached. Given this level of variation, we might expect potential errors in calculated plume heights when applying the meteorological observations to plume rise at the stack locations, in turn suggesting that a re-examination of plume rise using the model results is worthwhile."*

2. The model performance is not only influenced by the aspects forming the focus of this paper, but also by the accuracy of the meteorological part of the model, and by the numerics of

transport, notably the vertical diffusion and the handling of the point sources in the Eulerian framework. Their role is discussed only at the very end and, in my opinion, not sufficently in depth. In order to evaluate specific model aspects, one first needs to understand the performance of the model in general, with its strengths and weaknesses.

*The focus of this work is not to evaluate the overall model performance, but to evaluate how specific updates to the representation of the aerosol size distribution and the plumerise algorithm contribute to better model performance when compared to available observations. An evaluation of each aspect of a complex reaction-transport model is beyond the scope of a single paper. Nevertheless, as we already stated in the discussion section of the manuscript, we have carried out a sensitivity run which showed that variations in the magnitude of model diffusivity had a minimal impact the predicted plume behaviour and on the vertical distribution of $SO_2$ plumes at the point of release from the stacks, though we acknowledge that the model's tendency to overpredict the rate of decrease of air temperature with height may influence the shape of the diffusivity profile. We have also added additional references on the description and evaluation of the vertical diffusion scheme used in the meteorological portion of the model (Mailhot and Benoit 1982), as well as a more recent publication on the overall description and performance of the meteorological model as a whole (Girard et al., 2014), in Table 1 of the revised manuscript.*

3. The statistical approach chosen for the evaluation of the model options relies on metrics which exclusively are based on "match in time and space" data pairs. It is well known in air-pollution modelling that for near-source conditions (which is what we find here), there is often too much "noise" in the data (be it due to the stochastic nature of the plume, be it due to unresolved meteorological variability) to give meaningful results. Correspondingly, some of the statistical parameters are not very good. Therefore, global comparisons (such as deviations from the cumulative frequency distribution, statistics of cross-wind integrated values, or average dependency on key parameters such as stability and wind speed) are often used to assess models in a more robust way.

*We have added a simple table (Table 7, revised manuscript) showing the frequency distribution of the predicted versus observed plume rise from the three different variations on plume rise examined here. This new table is in agreement with the measurement statistics in that both show that the layered approach provides a better fit to observations, with a distribution more centered around the model:observation 1:1 line. While we agree with the reviewer regarding the difficulties associated with use of matching pairs for near-source conditions, we nevertheless respectfully hold that improvements in these statistics represent real improvements in model performance. For example, while a mean bias score is the average deviation between model and observed pairs, this average is over a large set of conditions, hence should be subject to less issues associated with the stochastic nature of the plume on any given hour. While near-source comparisons are often difficult due to the nature of the near-source region (as the reviewer suggests), improvements in these statistics nevertheless imply real improvements in model performance.*

4. The paper is written well on the "small scale" (apart from numerous technical deficiencies as listed below), but the broad topics could be worked out more clearly. In the end, the findings are: twelve aerosol size bins are better than two (not surprising, but good to see it quantified), there is

an improvement by using the model's vertical profile information for plume rise calculation but given the model's deficiencies the overall conclusion seems to be not so clear, and no improvement was found for using hourly stack data, but it remains unresolved why. We may wonder whether the work is mature enough for publication if we consider this state of the quintessential findings.

*We note that the results presented in the work show that the use of stack-location-specific meteorological information combined with the residual buoyancy calculations provides a considerably more accurate estimate of plume rise than the top-of-stack stability parameterizations often used in air-quality models. We have provided an additional table which shows that the distribution of plumes is better represented with the residual buoyancy calculation than with the top-of-stack stability plume rise calculation. The average plume rise calculated using the CEM-based data is closer to the observations than the annual totals, from both the original analysis and the additional table. While both the CEMS (using the hourly stack data) and the non-CEMS model scenarios are very close, the key point is that the revised, residual buoyancy plume rise algorithm has much better performance than the original algorithm. We have noted in the revised manuscript that "the relatively small differences between Figure 8(b) and 8(c), and between the last two columns of Table 7, imply that the residual buoyancy approach of equations 9 was relatively insensitive to the range of the initial buoyancy flux resulting from the two sets of emissions data used here, compared to the temperature gradients in equation (5)." We have also mentioned the insensitivity of the residual buoyancy calculation to the range of initial buoyancy flux in the revised conclusions; "However, the latter approach was also shown to be relatively insensitive to the range of initial buoyancy fluxes resulting from the two different emissions estimates, with the use of hourly observed (and presumed more accurate) stack parameters resulted in a slight degradation of performance relative to the use of annual reported values for these parameters."*

*Both sources of emissions data are limited by the model resolution and the independent verification of the accuracies of either is not available. We also point out in the revised manuscript that both sources of data have inherent errors. For example, as mentioned in the emissions paper referenced by this work (Zhang et al., 2018), and clarified/noted in the revised manuscript, the data referred to as CEMS here also contains engineering estimates of "upset conditions" wherein facility emissions are redirected to a flare stack for which direct emissions observations are not possible. That is, what we have referred to as "CEMS" data incorporates considerable associated uncertainties – this should have been included in the original manuscript and not left as a reference to the emissions paper alone. This has been mentioned in the description of the CEMS data as follows, "…and second with emissions information derived from a combination of CEMS hourly stack parameters as well as engineering estimates of emissions during "upset conditions" in which the effluent is redirected to flare stacks (the latter estimates are considerably more uncertain than the CEMS information, but are nevertheless included here since they result in substantial changes in pollutant emissions and plume characteristics, see Zhang et al, 2018)."*

**RC2: Specific Comments**

1. Page 2, l. 18: Why are you thinking that reasons for weak performance include only those meteorological variables that are used for the plume rise calculation, but not, for example, wind direction?

*We acknowledge that the model's performance is of course the result of many factors. The intent of our work is rather to evaluate the relative impact of the plume rise calculations on the results. The given sentence has been modified to "(iii) errors in meteorological forecast variables (including wind speed and direction, etc., as well as those used in calculating plume rise)".*

2. The model overview section lacks information on the numerical scheme used for vertical diffusion even though this is crucial in the context of study (cf. discussion on p. 24). The main reference for the MACH model seems to be Makar et al. (2010) – an extended abstract that would not be available for most people who haven't attended the conference as it is not freely accessible. Is there no more detailed and open description of this model? Note that also the Coté et al. citation is one of those for which the reference is missing. In addition, the handling of the point sources is not described (usually, Eulerian models use some sub-model to track plumes until they match the size of the grid cells).

*The revised manuscript includes a new Table (Table 1) which gives the main references for the meteorological (GEM) components for the model, including the reference for the Moist TKE approach used for calculating vertical diffusivity. Regarding the online air quality GEM-MACH model, the first overall description reference is Moran et al. (2010) (and not Makar et al. (2010) as stated by the reviewer), and hence we feel obliged to include it in published descriptions of GEM-MACH. However, this is not the only description of the model or its components, and others from the journal literature appeared in the original manuscript; we cited Gong et al (2003) for the aerosol microphysics, Gong et al (2006) for the aqueous-phase chemistry, Makar et al, (2003) for the inorganic heterogeneous chemistry, Lurmann et al. (1986) and Stroud et al. (2008) for the gas-phase mechanism , Zhang et al. (2001, 2002, 2003) and Makar et al (2018) for the gas and particle deposition. We also cited the Air Quality Model Evaluation International Initiative papers Im et al (2015a,b) and Makar et al (2015(a,b)) papers, which contain detailed descriptions of the model, its chemical and physical parameterizations, and its performance relative to other models of its type. The revised reference list has been double-checked to include all references (including the papers by Côté et al).*

*While some air-quality models include a form of "plume in grid" parameterization which track emitted puffs in a Lagrangian sense or employ a Gaussian dispersion model at the sub-grid scale, these approaches have not become predominant for three main reasons: (1) they ultimately rely on the driving large-scale meteorology (which may be inaccurate, as already pointed out in our work and by the reviewer, reducing their potential advantages); (2) they may add considerably to the processing time (particularly if a large number of point sources, chemical reactions and multiple species are considered), and (3) most models employ a self-nesting capability which allow the models to locally go to higher resolution, negating some of the advantages to a plume-in-grid approach. Consequently, most air-quality models have continued to rely on the handling of point sources using a combination of plume rise algorithms, and nesting to higher resolutions, as has been done in our work.*

3. The model set-up description in section 2.2 is not easy to follow. It might be helpful to move some of the information into a table and to shorten the text.

*A table summarizing the model description has been added as suggested.*

4. Page 8, line 1: The plume's buoyancy flux is **not** dependent on the stack height (at least not directly).

*This typographical error has been corrected.*

5. From the sentence beginning on p. 11, l. 7, on, the text does not really belong to the section 2.2.3. It should become a section of its own, as it introduces the simulations forming the base of the rest of the paper (maybe merge with some parts of the 2.2 chapeau).

*Section 2.2.3 has been renamed "Sources of Emissions Data", and the remainder of the previous section 2.2.3 past the point noted by the reviewer has been split off and renamed "2.2.4 Simulation Scenarios"*

6. Page 11, Section 3.1: What is the justification for removing measurements with values exceeding some threshold? Without proper justification this would not be acceptable.

*The key phrase in the original manuscript was that "extreme single-hour measurements" have been removed. That is, if the time series jumps from a background value to something greater than 150ppbv ($SO_2$, $NO_2$, and $O_3$) or 150 $\mu g\ m^{-3}$ (PM2.5), then immediately back again in the next hour, that jump is assumed to be due to instrumentation error and/or calibration times in the measurement record. In contrast, a rise above these levels for more than a single hour is retained. This has been clarified in the revised manuscript, viz: "The observation data have been filtered to remove extreme single-hour measurements that are greater than 150ppbv for $SO_2$, $NO_2$, and $O_3$, and 150 $\mu g\ m^{-3}$ for $PM_{2.5}$ (single-hour spikes of this nature in hourly records are assumed to correspond to instrumentation errors or calibration times for the instruments)."*

7. Page 12, Section 3.2: The phrase 'spatial linearly interpolated model values at the models chemistry time resolution of 2 minutes' is awkward. If you have 10 s data as said before, why do you need to interpolate for obtaining 2 min data? Also, it would be good to know which distance corresponds to both 10 s and 2min flight data, and how this compares to the model's grid size.

*The portion of the sentence has been corrected to "linearly interpolated values in time and space from the model's 2 minute time step and 2.5km resolution". We have also added the sentence: "The nominal cruise speed of the National Research Council Convair 580 used in the experiment is 550 km/hour; a 10 second time interval thus represents an observation integration distance of 1.528 km, and a two minute time interval an observation integration distance of 18.3 km."*

8. Section 4 (Results and Discussion) needs to be structured into subsections.

*This has been done, with subsections 4.1 Spatial Heterogeneity of Meteorological Conditions, 4.2 Two-bin versus Twelve-bin Evaluation, and 4.3 Plume Rise Algorithm Evaluation.*

9. Table 1: Apart from widely used or self-explanatory metrics such as FAC2, RMSE or r , the metrics parameters need to be defined.

*We have included a new Table; Table 2, which is now referenced at the start of section 3, and includes the mathematical definitions of all of the metrics.*

10. Page 12, l. 31: "Figure 2 shows that the model simulations are biased high for particles less than 5 μm diameter, and biased low for the larger particle sizes." As this figure only shows results for PM2.5, a statement on larger particles can't be based on it

*This was a typo; the text should have referred to 'concentration' and not 'particle sizes'; this has been corrected in the updated manuscript.*
.
11. Page 13, l. 14: Information on the bin sizes belongs to the model description section, not the result section.

*We have added a description of the cut sizes for both 2 and 12 bin simulations to the model description section. However, the mention of the 2 bin cut sizes is necessary here to explain why a comparison between the 2-bin model results with the aircraft observations is not appropriate (the 2-bin model lacks the size cut resolution to be able to simulate the PM1 observed by the aerosol mass spectrometer aboard the aircraft). The sentence has been changed to "The aircraft's AMS instrument measures speciated atmospheric particle concentrations for particles less than 1μm size, and therefore cannot be compared with the 2-bin model results because the smaller size bin (with upper diameter size cut 2.56 μm) will be biased high relative to the 1 μm size cut of the AMS."*

12. Page 13: The second paragraph on this page contains a number of statements about results without pointing to the figures or tables which show them.

*The revised manuscript references Table 4 for this paragraph.*

13. Concerning the model performance for PM, it should be discussed that even though the twelve-bin version leads to significant improvements, major discrepancies to observations remain.

*The last sentence of the new section 4.2 has been modified to read: "The use of the 12-bin size distribution (purple histogram bars, Figure 3) improves the fit to the observations (blue histogram bars), in comparison to the 2-bin distribution results (red histogram bars), though significant over-predictions of the frequency of low concentration events and under-prediction of high concentration events, remain.".*

14. A number of tables are presented where several metrics are used to compare various model versions, with the best one being emphasised by bold print. Sometimes, differences are tiny and probably insignificant. Only those values that are significantly better should be highlighted to

avoid a wrong impression of the results (for example, in Table 3 the model version seems to have no impact for O3 but we get the impression that the simpler model is better.)

*The problem with this suggestion is that different readers may have different ideas regarding what is considered a 'significant' change, what is considered to be a 'tiny' difference, etc. We feel that the readers will look at both the numbers themselves as well as the highlighting, as the reviewer did, to note the relative level of differences. We mentioned some of these relative differences in the original text as well, as a caution to the reader not to base their judgement on the highlighting alone, e.g., with reference to the ozone evaluation in Table 3: "Ozone, in contrast, is created or destroyed through secondary chemistry over relatively longer time-spans than the transport time from the sources in this comparison (spatial scales on the order of 10's of km). Accordingly, the impact of the plume rise of $NO_x$ on ozone formation is relatively minor, usually in the third decimal place (though first decimal place improvements occur for the mean bias with the use of the new plume rise algorithm)."*

15. Why is the use of hourly emission data beneficial for NO2 but detrimental for SO2?

*One significant difference between $SO_2$ and $NO_2$ for the study region is that the latter originates almost completely in major point sources, while only 40% of the latter originates in major point sources, the rest in area sources (heavy hauler fleets used by the open pit mine operators). The $NO_2$ values will thus be due to a combination of sources, with the possibility of compensating errors at the emissions level influencing the net model $NO_2$ concentration .*

16. The discussion paper does not comply with the ACP Data Policy; it does not have a "Data availability" section and says nothing about data availability

*This has been added to the revised manuscript (in our other papers with ACP, this has come as a request from the Editor following the completion of the review process, sorry for not having added it to the submission): all of the data used here are publicly available on the oil sands data archive or the website of the Wood Buffalo Environmental Association. We have also added the standard Author Contributions, Competing Interests, Special Issue Statement, and Acknowledgements, to the revised manuscript.*

**A chemical transport model study of plume rise and particle size distribution for the Athabasca oil sands**

Ayodeji Akingunola[1], Paul. A. Makar[1], Junhua Zhang[1], Andrea Darlington[2], Shao-Meng Li[2], Mark Gordon[3], Michael D. Moran[1], Qiong Zheng[1]

[1]Modelling and Integration Section, Air Quality Research Division, Environment and Climate Change Canada
[2] Processes Research Section, Air Quality Research Division, Environment and Climate Change Canada
[3]Centre for Research In Earth And Space Engineering, York University, Toronto Canada

*Correspondence to*: Ayodeji Akingunola (Ayodeji.akingunola@canada.ca),  Paul Makar (paul.makar@canada.ca)

**Abstract.**

We evaluate four high-resolution model simulations of pollutant emissions, chemical transformation and downwind transport for the Athabasca oil sands using the Global Environmental Multiscale – Modelling Air-quality and Chemistry (GEM-MACH) model, and compare model results with using surface monitoring network and aircraft observations of multiple pollutants, for simulations spanning a time period corresponding to an aircraft measurement campaign in the region in the summer of 2013.  We have focussed here on the impact of different representations of the model's aerosol size distribution and plume-rise parameterization on model results.

The use of a more finely resolved representation of the aerosol size distribution was found to have a significant impact on model performance, reducing the magnitude of the original surface PM2.5 negative biases by 32%.32%, from -2.62 to -1.72 $\mu g\ m^{-3}$.

We compared model predictions of $SO_2$, $NO_2$, and speciated particulate matter concentrations from simulations employing the commonly-used Briggs (1984) plume-rise algorithms to redistribute emissions from large stacks. with stack plume observations. As in our companion paper (Gordon *et al*., 2018), we found that Briggs algorithms based on estimates of atmospheric stability at the stack height resulted in under-predictions of plume rise, with 116 out of 176 test cases falling below the model:observation 1:2 line, 59 cases falling within a factor of 2 of the observed plume heights, and an average model plume height of 289 m compared to an average observed plume height of 822 m.  We used a high resolution meteorological model to confirm the presence of significant horizontal heterogeneity in the local meteorological conditions driving plume rise.  Using these simulated meteorological conditions at the stack locations, we found that a layered buoyancy approach for estimating plume rise in stable to neutral atmospheres, coupled with the assumption of free rise in convectively unstable atmospheres, resulted in much better model performance relative to observations (124 out of 176 cases falling within a factor of two of the observed plume height, with 69 of these cases above and 55 of these cases below the 1:1 line and within a factor of two of observed values).  This is in contrast to our companion paper, wherein this layered approach (driven by meteorological observations not co-located with the stacks) showed a relatively modest impact on predicted plume heights.  As in our companion paper (Gordon *et al*., 2018), we found these algorithms resulted in under-predictions of plume rise, with 39 to 60% of predicted plume heights falling below half of the observed plume heights.  However, we found here that a layered buoyancy approach for stable to neutral atmospheres, coupled with the assumption of free rise in convectively unstable atmospheres, resulted in much better model performance, both for atmospheric constituent concentrations and the predicted height of the plumes.  Persistent issues with over-fumigation of

plumes in the model were linked to a more rapid decrease in simulated temperature with increasing height than was observed. This in turn may have led to overestimates of near-surface diffusivity, resulting in excessive fumigation.

**1 Introduction**

[revised manuscript text omitted]

Formatted Table

[revised manuscript text omitted]

$$F_{j+1} = \begin{cases} F_j - 0.015 s_j F_{j-1}^{\frac{1}{3}}(z_{j+1}^{\frac{8}{3}} - z_j^{\frac{8}{3}}), & \text{vertical plumes} \\ F_j - 0.053 s_j U_j (z_{j+1}^{3} - z_j^{3}), & \text{bent plumes} \end{cases} \qquad (9)$$

For vertical plumes,

$$F_{j+1} = F_j - 0.015 s_j F_{j-1}^{\frac{1}{3}} \left( z_{j+1}^{\frac{8}{3}} - z_j^{\frac{8}{3}} \right),  \qquad (9a)$$

and for bent plumes

$$F_{j+1} = F_j - 0.053 s_j U_j \left( z_{j+1}^{3} - z_j^{3} \right),  \qquad (9b)$$

Here, $s_j$ is the local stability parameter for a given layer, calculated using (5) and layer-specific temperature values, and $z_j$ is the plume rise height when the plume reaches the bottom of the model's -j'th layer. Briggs (1984) recommended the use of *both* formulae of (9), with the formula with greatest decrease in flux being used as the final value. Briggs also noted that the transition to bent plumes happens at a relatively low height above the stack, implying that that the residual buoyancy between layers is lost faster under windy conditions. At the stack height, $F_{j=0} = F_b$, and $z_j = 0$ (that is, the vertical distances are relative to the top of the stack). When the residual buoyancy flux becomes negative in (9), indicating that the plume height has been surpassed, the calculation is repeated to find the value of $z$ for which $F = 0$; the sum of this and the layer thicknesses transitioned to this height becomes the predicted plume rise. In our companion paper (Gordon *et al.*, 2018), this approach was found to provide similar results to the original Briggs' algorithms when driven by observations not co-located with the stacks. Our work here indicates that this algorithm has the potential to provide a more accurate estimate of plume rise, subject to caveats described below.

We note that the numerical coefficients in (9); 0.015 and 0.053, stem from two parameters; the entrainment constant for vertical rise conditions ($\alpha$, the entrainment coefficient for vertical plumes, nominally set to 0.08 by Briggs based on observations published in 1975 - the parameter in the first equation of (9) is a non-linear function of this $\alpha$ term; and $\beta'$, the entrainment coefficient internal radius for bent-over plumes, set by Briggs to 0.4, though ranges from 0.45 to 0.52 were quoted elsewhere in Briggs, 1984). The choice of these parameters are based on data which are now over 40 years old, and may present an opportunity for future improvement of this revised plume rise approach.

The above formula (9) was recommended by Briggs for conditions which are stable to neutral at the stack height. We have defined stability in this case by comparing the dry adiabatic lapse rate to the local temperature lapse rate predicted by the model at the stack height and above. Briggs (1984) provided no equivalent formula for unstable conditions at the stack height, followed by stable profiles at higher elevations. The approach taken here has been to assume under convectively unstable conditions, the plume rises without loss of energy (that is, an assumption of zero entrainment) until the predicted temperature profile once again falls below the dry adiabatic lapse rate. Our first order approximation is thus to assume that under unstable conditions, there is minimal mixing entrainment of the rising plume with the surrounding atmosphere. This approach differs from that of Turner (1991), and the layered approach described in Byun and Ching (1999) where the residual buoyancy flux between layers is determined using different formulae based on the model-determined local atmospheric stability.

As in the original algorithm, the plume top and plume bottom are evaluated using equation (8) after the final plume rise has been evaluated. We do not apply the penetration equations (Eq.6 and Eq.7) since these corrections should be unnecessary in an approach making use of local changes in residual buoyancy. In our companion paper, this algorithm is referred to as the "layered approach".

**2.2.3 Hourly Emission Stack Temperature and Volume Flow Rate**

We turn next to the available emissions data for driving the plume rise algorithms. Under Canadian federal reporting requirements to the National Pollutant Release Inventory (NPRI), annual total emissions of $SO_2$ and $NO_x$ from facilities are reported, along with a single set of stack parameters (stack height, stack diameters, average exit temperature, and average exit

velocity) to represent emissions throughout the year. In addition, hourly Continuous Emissions Monitoring data from large stacks are reported to the government of Alberta. These data include the hourly mass of emissions of $SO_2$ and $NO_2$, as well as hourly estimates of the time-varying stack parameters (volume flow rates and temperatures).

**2.2.4 Simulation Scenarios**

Our first two simulations use the "standard" annual NPRI reported stack parameters and the original plume rise algorithm for the 2-bin and 12-bin aerosol size distributions, while our second two simulations use the modified plume rise algorithm, first with the NPRI stack parameters, and second with emissions information derived from a combination of CEMS hourly stack parameters as well as engineering estimates of emissions during "upset conditions" in which the effluent is redirected to flare stacks (the latter estimates are considerably more uncertain than the CEMS information, but are nevertheless included here since they result in substantial changes in pollutant emissions and plume characteristics, see Zhang et al, 2018). and second with CEMS-derived hourly stack parameters. The four scenarios examined are thus:

(1) A "2-bin" simulation: NPRI stack parameters, 2-bin aerosol size distribution, and the original plume rise
(2) A "12-bin" simulation: As in (1), but employing the 12-bin aerosol size distribution. Differences between (1) and (2) thus show the impact of the aerosol size distribution on performance.
(3) A "Plume rise" simulation: employing the layered plume rise algorithm, with emissions as in (2) Differences between (2) and (3) thus show the impact of the revised plume rise algorithm alone.
(4) An "Hourly" simulation: employing the layered plume rise algorithm, with volume flow rates and temperatures taken from the hourly CEMS data along with upset conditions. Differences between (3) and (4) thus show the impact of the initial buoyancy flux on the resulting plume rise, using the revised algorithm.

All of these simulations make use of the CEMS-derived mass of emitted $SO_2$ and $NO_x$.

**3 Observations**

The comparative statistics presented through this study were computed using the 'modstat'modStat' function from the openair'openair' R package (Carslaw and Ropkins, 2012), for complete pairs of valid model and observation data. The set of statistical measures and their formulas are presented in Table 2. Both surface monitoring network and aircraft observations have been used for model evaluation.

**3.1 WBEA Surface Monitoring Networks**

For the purpose of model evaluation, we have used hourly measurements of surface concentrations of PM2.5, $SO_2$, $NO_2$, and $O_3$ from a network of 10 air quality monitoring stations in the province of Alberta managed by the Wood Buffalo Environmental Association (WBEA) (see Figure 1(dupper-left panel)). The observation data have been filtered to remove extreme single-hour measurements that are greater than 150ppbv for $SO_2$, $NO_2$, and $O_3$, and 150 µg m$^{-3}$ for PM2.5 (single-hour spikes of this nature in hourly records are assumed to correspond to instrumentation errors or calibration times for the instruments). The observation data have been filtered to remove extreme single-hour measurements that are greater than 150ppbv for $SO_2$, $NO_2$, and $O_3$, and 150 µg m$^{-3}$ for PM2.5. Observations from August 10th, 2013 to September 10th, 2013 were selected for comparison to the model results, to align to with the period covered by the JOSM 2013 intensive aircraft measurement campaign.

**3.2 JOSM Summer 2013 Intensive Campaign**

From August 10th to September 10th, 2013, the National Research Council of Canada Convair aircraft was used as a mobile measurement platform to sample atmospheric constituents in the region of the Athabasca oil sands, with twenty-two flights taking place during the given time period (Figure 1, lower-left panel).  These flights included flight paths designed for emission estimation, for the study of downwind transport and chemical transformation, and for satellite validation. Emission estimation flights took place around individual facilities at multiple altitudes, with the concentration and meteorological information gathered subsequently used to estimate fluxes entering and leaving the facility, and hence estimate emissions directly from aircraft observations (Gordon *et al*., 2015; Li *et al.*, 2017)  Transformation flights were designed to follow plumes downwind, with observations taken in cross-sections at set distances downwind perpendicular to the plume direction, in order to study chemical transformations between point of emission and downwind receptors (cf. Liggio *et al*., 2016).  Satellite validation flights incorporated aircraft vertical spirals at satellite overpass times, in order to improve satellite data retrieval algorithms (Whaley *et al*., 2018; Sheppard *et al.*, 2015).  Here, we compare model predictions for our different simulations for $SO_2$, $NO_2$ and for PM1 sulfate, ammonium, and total organics to observations taken on-board the Convair using TS43, TS42 and Aerodyne Aerosol Mass Spectrometers (AMS) instruments, respectively. In order to allow for comparisons to the results from GEM-MACHv2 2.5km oil sands model domain simulations, 10-second averages of the aircraft's positional data (latitude, longitude, elevation, and time) were created for all 22 flights. These data were in turn used to extract the corresponding linearly interpolated values in time and space from the model's 2 minute time step and 2.5km resolution, for each of the  species observed aboard the aircraft that were used for the model comparison. The nominal cruise speed of the National Research Council Convair 580 used in the experiment is 550 km/hour; a 10 second time interval thus represents an observation integration distance of 1.528 km, and a two minute time interval an observation integration distance of 18.3 km

**4 Results and Discussion**

**4.1  Spatial Heterogeneity of Meteorological Conditions**

We noted in our companion paper (Gordon *et al.*, 2018) that meteorological observations varied substantially in the study region depending on location, citing this as a possible confounding factor on the results of tests of the plume rise algorithms.  This spatial heterogeneity was well captured by the high resolution GEM-MACH simulations, as is demonstrated by the example depicted in Figure 2,  which shows the typical local variation in planetary boundary layer height (Figure 2(a)), ranging from about 1200m to 400m, the lower values corresponding to the main cleared areas (open pit mines, settling ponds) of the industrial facilities.  The corresponding temperature profiles in several locations marked in Figure 2(a) are given in Figure 2(b):  These show a substantial difference in model predicted stability at the three meteorological observation locations of Gordon *et al.* (2018) (windRASS, AMS03, and AMS05), and substantial differences between these and the locations of the main stacks of some of the facilities (Syncrude 1, CNRL, and Suncor).  The temperature profiles show that the height and strength of the inversion may vary by over 100m in the vertical, and that the profiles do not merge with the larger scale flow until an elevation of 750m asl (450m agl) is reached.  Given this level of variation, we might expect potential errors in calculated plume heights when applying the meteorological observations to plume rise at the stack locations, in turn suggesting that a re-examination of plume rise using the model results is worthwhile.

[Figure]

Figure 2: Examination of meteorological heterogeneity in the study area. (a) PBL heights (locations of large emitting stacks shown as circles, meteorological observation sites as stars). (b) Predicted temperature profiles at these locations and times; symbols indicate locations of model levels (a terrain following coordinate system is used in GEM-MACH).

**4.2 Two-bin versus Twelve-bin Evaluation**

[revised manuscript text omitted]

Table 1

**Table 2**: Statistical measures used in comparing model results with observations.

| Statistic | Formula |
|---|---|
| | $M_i$ = model time series; $O_i$ = observation time series |
| Number of complete data pair | $n$ |
| Fraction of predictions within a factor of two | $FAC2 = 0.5 \le \dfrac{M_i}{O_i} \le 2.0$ |
| Mean bias | $MB = \dfrac{1}{n}\sum_{i=1}^{N} M_i - O_i$ |
| Mean Gross Error | $MGE = \dfrac{1}{n}\sum_{i=1}^{N} |M_i - O_i|$ |
| Normalised mean bias | $NMB = \dfrac{\sum_{i=1}^{n} M_i - O_i}{\sum_{i=1}^{n} O_i}$ |
| Normalised mean gross error | $NMGE = \dfrac{\sum_{i=1}^{n}|M_i - O_i|}{\sum_{i=1}^{n} O_i}$ |
| Root mean squared error | $RMSE = \left(\dfrac{\sum_{i=1}^{n}(M_i - O_i)^2}{n}\right)$ |
| Correlation coefficient | $r = \dfrac{1}{(n-1)}\sum_{i=1}^{n}\left(\dfrac{M_i - \bar{M}}{\sigma_M}\right)\left(\dfrac{O_i - \bar{O}}{\sigma_O}\right)$ |
| Coefficient of Efficiency | $COE = 1.0 - \dfrac{\sum_{i=1}^{n}|M_i - O_i|}{\sum_{i=1}^{n}|O_i - \bar{O}|}$ |
| Index of Agreement | $IOA = \begin{cases} 1.0 - \dfrac{\sum_{i=1}^{n}|M_i - O_i|}{c\sum_{i=1}^{n}|O_i - \bar{O}|}, & \text{when } \sum_{i=1}^{n}|M_i - O_i| \le c\sum_{i=1}^{n}|O_i - \bar{O}| \\[2ex] \dfrac{c\sum_{i=1}^{n}|O_i - \bar{O}|}{\sum_{i=1}^{n}|M_i - O_i|} - 1.0, & \text{when } \sum_{i=1}^{n}|M_i - O_i| > c\sum_{i=1}^{n}|O_i - \bar{O}| \end{cases}$ |

[revised manuscript text omitted]
}\left(T_{s,r} - T_a\right)}{V_o T_{s,r}\left(T_{s,o} - T_a\right)} \qquad (10)$$

Where the subscripts $r$ and $o$ indicate the annual reported and hourly observed values of each quantity. Assuming an ambient

10  temperature at stack height of 291K, the value of R is 2.28; that is, the initial buoyancy flux of the Plume rise simulation is over double that of the Hourly simulation. The hourly values are k̶n̶o̶w̶n̶ believed to be more realistic during the period simulated (though include engineering emissions estimates during upset conditions) – the revised algorithm, while providing better results than the original, thus still has a tendency to under-predict the plume heights. In our companion paper, we found that the revised algorithm (therein referred to as the "layered approach") had no significant advantages over the original Briggs algorithms – here

15  we have found this revised approach has considerable benefit, while showing the same overall tendency to under-predict plume heights as in our companion paper.

[Figure]

[Figure]

Figure 7.  Zoomed-view of Figure 5.  (a) 17:42-17:54 UTC, observations interpolated from successive flight passes between 17:00 and 18:19 UTC.  (b) 19:08-19:20 UTC, observations interpolated from successive flight passes between 18:42 and 19:45 UTC

In order to demonstrate the extent to which the plume rise values themselves differ between flights, we have compared the calculated plume heights from each of the three algorithms examined here for 8 stacks (located at the Syncrude, Suncor, and CNRL facilities) against observations during the course of the study (Figure 78).  The observed plume rise values here were derived from estimates of the $SO_2$ plume centres from the aircraft campaign's emission box flights as estimated in our companion paper (Gordon *et al*. (2018)). Despite the differences visible in Figures 56 and 67, for flight 12, Figure 78 shows that the revised algorithm has a significant impact on calculated plume heights, greatly increasing the number falling within a factor of two of the observations (>70%), while the original algorithm has the majority of calculated plume heights falling below the 1:1 line, in accord with Gordon *et al*. (2018)).  However, the impact of the differences in volume flow rates and temperatures (Figure 78(b) versus Figure 78(c))  are usually relatively minor, with the exception of a few additional points falling below the 1:2 line for the Hourly (Figure 78(c)) simulation.  Table 7 shows the relative distribution of the 176 test cases compared in terms of their distribution about the 1:2, 1:1 and 2:1 lines of the scatterplots of Figure 8.  The revised plumerise approach results in a

significant improvement in the distribution, and the use of CEMS data results in a slight further improvement in the average predicted plume height. We note that the relatively small differences between Figure 8(b) and 8(c), and between the last two columns of Table 7, imply that the residual buoyancy approach of equations 9 was relatively insensitive to the range of the initial buoyancy flux resulting from the two sets of emissions data used here, compared to the temperature gradients in equation (5).

The large deviation between the annual reported and measured stack parameters for flight 12 may thus be an anomaly relative to the entire record across all 8 stacks examined here. Nevertheless, Figures 34 to 67 suggest that all of model simulations have a tendency to overestimate fumigation, so we continued our examination using Flight 12 as a case study.

The model concentrations of primary pollutants are also modified by vertical diffusion and advection. The use of a plume rise algorithm simultaneously with vertical diffusion implies the potential for "double-counting" of some proportion of the vertical mixing, in that the observation-based plume rise algorithms *de facto* incorporate vertical diffusion in their estimates of plume rise, while air-quality models must apply diffusion at all model grid-squares, including those in which plume rise algorithms have already distributed emitted mass in the vertical. If the relative impact of vertical diffusion versus buoyant plume rise is strong, this may result in excessive vertical mixing; the model effectively "double-counting" the vertical diffusion component of the net rise. The potential for overestimates of model diffusivity magnitudes resulting in excessive vertical mixing to the ground was investigated by carrying out a sensitivity run for Flight 12 in which diffusivities in the column were halved prior to their use in calculating vertical diffusion. This sensitivity run showed a minimal impact on model results – the magnitude alone of vertical diffusion did not influence the fumigation noted below. However, this test did not examine the potential changes associated with different magnitude changes in diffusivity as a function of height.

[Figure]

Figure 78: Observed plume rise heights during aircraft emission box flights compared to model calculated plume rise using; (a) the original plume rise algorithm; (b) new plume rise algorithm; and (c) new plume rise algorithm and CEMS hourly stack temperature and volume flow rate.

**Table 7**: Comparison of model plume rise performance

| Number (model:observed)
Total number of comparisons: 176 | Original Plume Rise Algorithm | New Plume Rise Algorithm (Layered Approach) | New Plume Rise Algorithm driven by CEMS data |
|---|---|---|---|
| Below 1:2 line | 116 | 24 | 29 |
| Between 1:2 and 1:1 line | 44 | 54 | 55 |
| Between 1:1 and 2:1 line | 15 | 75 | 69 |
| Above 2:1 line | 10 | 22 | 22 |
| Average plume height (m) (Observed: 822 m) | 289 | 935 | 914 |
| Ratio of average simulated to observed plume height | 0.35 | 1.13 | 1.11 |

[revised manuscript text omitted]
 latter approach was also shown to be relatively insensitive to the range of initial buoyancy fluxes resulting from the two different emissions estimates, with the use of hourly observed (and presumed more accurate) stack parameters resulted in a slight degradation of performance relative to the use of annual reported values for these

20  parameters.  Further investigation using a specific case study suggested that the improvements associated with the revised algorithm may in part be due to model positive biases in lower atmospheric temperature, resulting in model underestimates in the magnitude of atmospheric temperature gradients.  Nevertheless, the revised approach was found to correct much of the predominantly negative bias in predicted plume height seen for Briggs' original algorithms, correcting the biases in plume height noted in our companion paper, in which the algorithms were driven using observed meteorology.

25  Despite these improvements, and the tendency of the model to underestimate temperature gradients, the model still over-predicts the extent of fumigation for all plume rise algorithms tested, implying the need for further work.  The revised approach found to be the most favorable in the current work is based on two key parameters; entrainment coefficients determined by Briggs from data collected in 1975 to be approximately 0.08 and 0.4 respectively; we recommend that these coefficients be re-estimated using more recent data.

30  Our simulations have shown that the choice of a plume rise parameterization has a very significant impact on downwind concentrations of $SO_2$ from the oil sands sources, with the approaches having the more accurate plume heights also resulting in significant reductions in surface $SO_2$, and increases in $SO_2$ aloft, helping to correct pre-existing positive and negative biases in the model at these elevations.  Smaller impacts were found for $NO_2$, and minimal impacts for ozone.

*Code and data availability*

35  The aircraft observations used in this study are publicly available on the ECCC data portal (https://www.canada.ca/en/environment-climate-change/services/oil-sands-monitoring/monitoring-air-quality-alberta-oil-sands.html). The hourly surface monitoring network data are from the public website of the Wood Buffalo Environmental Monitoring Association (http://www.wbea.org/network-and-data/historical-monitoring-data).  The model results are available

upon request to Ayodeji.Akingunola (ayodeji.akingunola@canada.ca). GEM-MACH, the atmospheric chemistry library for the GEM numerical atmospheric model (© 2007–2013, Air Quality Research Division and National Prediction Operations division, Environment and Climate Change Canada), is a free software which can be redistributed and/or modified under the terms of the GNU Lesser General Public License as published by the Free Software Foundation – either version 2.1 of the license or any later version. The specific GEM-MACH version used in this work may be obtained on request to ayodeji.akingunola@canada.ca. Much of the emissions data used in our model are available online: Executive Summary, Joint Oil Sands Monitoring Program Emissions Inventory report (https://www.canada.ca/en/environment-climate-change/services/science-technology/publications/joint-oil-sands-monitoring-emissions-report.html; Joint oil sands monitoring program emissions inventory report, 2018) and Joint Oil Sands Emissions Inventory Database (http://ec.gc.ca/data_donnees/SSB-OSM_Air/Air/Emissions_inventory_files/) and more recent updates may be obtained by contacting Junhua Zhang or Mike Moran (junhua.zhang@canada.ca; mike.moran@canada.ca).

*Author Contributions:* AA and PAM were responsible for the study design and methodology, model simulations, comparison to observations, and the writing of the manuscript and modifications of same. JZ, MDM and QZ contributed emissions data used in the modelling. AD and S-ML contributed aircraft observation data used for model evaluation. MG contributed aircraft plume height analyses, information on the companion paper, and contributed to the text and revisions of the manuscript.

*Competing Interests*: The authors declare that they have no conflict of interest.

*Special issue statement*: This article is part of the special issue "Atmospheric emissions from oil sands development and their transport, transformation and deposition (ACP/AMT inter-journal SI)". It is not associated with a conference.

*Acknowledgments:* This project was jointly supported by the Climate Change and Air Quality Program of Environment and Climate Change Canada, Alberta Environment and Parks, and the Oil Sands Monitoring program. The figures in this work were created using a combination of Environment Canada and Climate Change software and the R open-source programming language (R Core Team, 2017).